# Human Perception-based Evaluation Criterion for Ultra-high Resolution Cell Membrane Segmentation

## Abstract

Computer vision technology is widely used in biological and medical data analysis and understanding. However, there are still two major bottlenecks in the field of cell membrane segmentation, which seriously hinder further research: lack of sufficient high-quality data and lack of suitable evaluation criteria. In order to solve these two problems, this paper first introduces an Ultra-high Resolution Image Segmentation dataset for the Cell membrane, called U-RISC, the largest annotated Electron Microscopy (EM) dataset for the Cell membrane with multiple iterative annotations and uncompressed high-resolution raw data. During the analysis process of the U-RISC, we found that the current popular segmentation evaluation criteria are inconsistent with human perception. This interesting phenomenon is confirmed by a subjective experiment involving twenty people. Furthermore, to resolve this inconsistency, we propose a new evaluation criterion called Perceptual Hausdorff Distance (PHD) to measure the quality of cell membrane segmentation results. Detailed performance comparison and discussion of classic segmentation methods along with two iterative manual annotation results under existing evaluation criteria and PHD is given.

## 1 Introduction

Electron Microscopy (EM) is a powerful tool to explore ultra-fine structures in biological tissues, which has been widely used in the research areas of medicine and biology ( ERLANDSON (2009); Curry et al. (2006); Harris et al. (2006)). In recent years, EM techniques have pioneered an emerging field called "Connectomics" (Lichtman et al. (2014)), which aims to scan and reconstruct the whole brain circuitry at the nanoscale. "Connectomics" has played a key role in several ambitious projects, including the BRAIN Initiative ( Insel et al. (2013)) and MICrONS ( Gleeson & Sawyer (2018)) in the U.S., Brain/MINDS in Japan ( Dando (2020)), and the China Brain Project ( Poo et al. (2016)). Because EM scans brain slices at the nanoscale, it produces massive images with ultra-high resolution and inevitably leads to the explosion of data. However, compared to the advances of EM, techniques of data analysis fall far behind. In particular, how to automatically extract information from massive raw data to reconstruct the circuitry map has growingly become the bottleneck of EM applications.

One critical step in automatic EM data analysis is Membrane segmentation. With the introduction of deep learning techniques, significant improvements have been achieved in several public available EM datasets ISBI 2012 and SNEMI3D ( ISBI 2012 (2012); ISBI 2013 (2013); Arganda-Carreras et al. (2015b); Lee et al. (2017)). One of the earliest works ( Ciresan et al. (2012) used a succession of max-pooling convolutional networks as a pixel classifier, which estimated the probability of a pixel is a membrane. Ronneberger et al. (2015) presented a U-Net structure with contracting paths, which captures multi-contextual information. Fully convolutional networks (FCNs) proposed by Long et al. (2015) led to a breakthrough in semantic segmentation. Follow-up works based on U-net and FCN structure ( Xie & Tu (2015); Drozdzal et al. (2016); Hu et al. (2018); Zhou et al. (2018); Chaurasia & Culurciello (2017); Yu et al. (2017); Chen et al. (2019b)) have also achieved outstanding results near-human performance.

Despite much progress that has been made in cell membrane segmentation for EM data thanks to deep learning, one risk to these popular and classic methods is that they might be "saturated" at the current datasets as their performance appear to be "exceedingly accurate" ( Lee et al. (2017)). How

can these classic deep learning based segmentation methods work on new EM datasets with higher resolution and perhaps more challenges? Moreover, how robust of these methods when they are compared with human performance on such EM images?

To expand the research of membrane segmentation on more comprehensive EM data, we first established a dataset "U-RISC" containing images with original resolution ($10000 \times 10000$ pixels, Fig. 1). To ensure the quality of annotation, it also costs us over 10,000 labor hours to label and double-check the data. To the best of our knowledge, U-RISC is the largest uncompressed annotated and EM dataset today. Next, we tested several classic deep learning based segmentation methods on U-RISC and compared the results to human performance. We found that the performance of these methods was much lower than that of the first annotation. To understand why human perception is better than the popular segmentation methods, we examined in detail the Membrane segmentation results by these popular segmentation methods. How to measure the similarity between two image segmentation results has been widely discussed ( Yeghiazaryan & Voiculescu (2018); Niessen et al. (2000); Veltkamp & Hagedoorn (2000); Lee et al. (2017)). Varduhi Yeghiazaryan ( Yeghiazaryan & Voiculescu (2018)) discussed the family of boundary overlap metrics for the evaluation of medical image segmentation. Veltkamp. etc ( Veltkamp & Hagedoorn (2000)) formulated and summed up the similarity measures in a more general condition. In some challenges, such as ISBI2012 ( Arganda-Carreras et al. (2015a)), they also considered multiple metrics like Rand score on both original images and thinned images. However, we found there was a certain inconsistency between current most popular evaluation criteria for segmentation(e.g. F1 score, IoU) and human perception: while some figures were rated significantly lower in F1 score or IoU, they were "perceived" better by humans (Fig. 4).

Such inconsistency motivated us to propose a human-perception based criterion, Perceptual Hausdorff Distance (PHD) to evaluate image qualities. Further, we set up a subjective experiment to collect human perception about membrane segmentation, and we found the PHD criteria is more consistent with human choices than traditional evaluation criteria. Finally, we found the current popular and classical segmentation methods need to be revisited with PHD criteria.

Overall, our contribution in this work lies mainly in the following two parts: (1) we established the largest, original image resolution-based EM dataset for training and testing; (2) we proposed a human-perception based evaluation criterion, PHD, and verified the superiority of PHD by subjective experiments. The dataset we contributed and the PHD criterion we proposed may help researchers to gain insights into the difference between human perception and conventional evaluation criteria, thus motivate the further design of the segmentation method to catch up with the human performance on original EM images.

## 2  U-RISC: ULTRA-HIGH RESOLUTION IMAGE SEGMENTATION DATASET FOR CELL MEMBRANE

Supervised learning methods rely heavily on high-quality datasets. To alleviate the lack of training data for cell membrane segmentation, we proposed an Ultra-high Resolution Image Segmentation dataset for Cell membrane, called U-RISC. The dataset was annotated upon RC1, a large scale retinal serial section transmission electron microscopic (ssTEM) dataset, publically available upon request and described in the work of Anderson et al. (2011). The original RC1 dataset is a 0.25mm diameter, 370 TEM slices volume, spanning the inner nuclear, inner plexiform, and ganglion cell layers, acquired at 2.18 nm/pixel across both axes and 70nm thickness in z-axis. From the 370 serial-section volume, we clipped out 120 images in the size of $10000 \times 10000$ pixels from randomly chosen sections. Then, we manually annotated the cell membranes in an iterative annotation-correction procedure. Since the human labeling process is very valuable for uncovering the human learning process, during the relabeling process, we reserved the intermediate results for public release. The U-RISC dataset will be released on `https://Anonymous.com` on acceptance.

### 2.1  COMPARISON WITH OTHER DATASETS

ISBI 2012 (Cardona et al. (2010)) published a set of 30 images for training, which were captured from the ventral nerve cord of a Drosophila first instar larva at a resolution of $4 \times 4 \times 50$ nm/pixel through ssTEM (Arganda-Carreras et al. (2015b); ISBI 2012 (2012)). Each image contains $512 \times 512$ pixels, spanning a realistic area of $2 \times 2$ $\mu$m approximately. In the challenge of SNEMI3D (Kasthuri

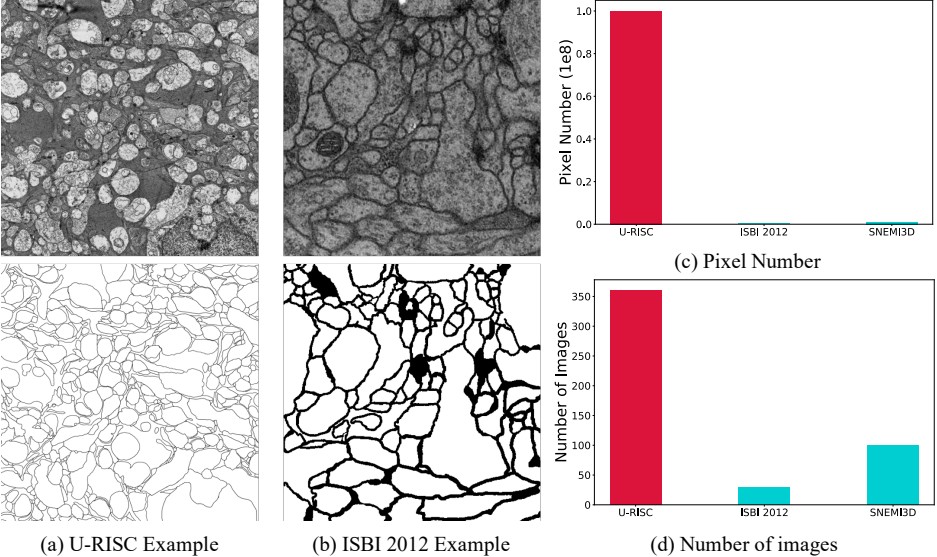

Figure 1: Dataset examples and statistical comparisons. (a) and (b) are examples of U-RISC and ISBI2012. The first line shows the original image, and the second line shows its annotation. (c) shows the pixel number of U-RISC, ISBI2012, and SNEMI3D datasets. (d) shows the image number of the three datasets.

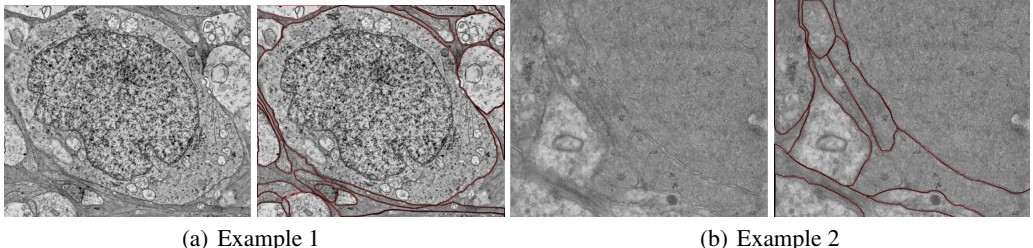

Figure 2: Examples of images with their labels. The left of (a) (b) are two original image parts, and the right are the images covered by their labels (red lines).

et al. (2015); ISBI 2013 (2013)), the training data is a 3D stack of 100 images in the size of $1024\times1024$ pixels with the voxel resolution of $6\times6\times29$ nm/pixel. The raw images were acquired at the resolution of $3\times3\times29$ nm/pixel using serial section scanning electron microscopy (ssSEM) from mouse somatosensory cortex (Kasthuri et al. (2015); ISBI 2013 (2013)). U-RISC contains 120 pieces of annotated images ($10000\times10000$ pixels) at the resolution of $2.18\times2.18\times70$ nm/pixel from rabbit retina.

Due to the difference of species and tissue, U-RISC can fill in the blank of annotated vertebrate retinal segmentation dataset. Besides that, U-RISC has some other characteristics which can be focused on in the future segmentation study. The first one is that the image size and realistic size of U-RISC is much larger, specifically, the image size of U-RISC is 400 and 100 times of ISBI2012 and SNEMI3D respectively, and the realistic size is 100 and 9 times of them respectively (Fig. 1 (c)), which can be applied in developing deep learning based segmentation methods according to various demands. And along with the iterative annotation procedure U-RISC actually contains 3 sets of annotation results with increasing accuracy, which could serve as ground truth at different level standard. And the total number of annotated images is 12 and 3.6 times of the public annotated images of ISBI2012 and SNEMI3D respectively (Fig. 1 (d)). An example of the image with its label is shown in the Supplementary. Due to the limitation of the size of the supplementary material, we only uploaded a quarter (5000 $\times$5000 pixels) size of the original image with its label.

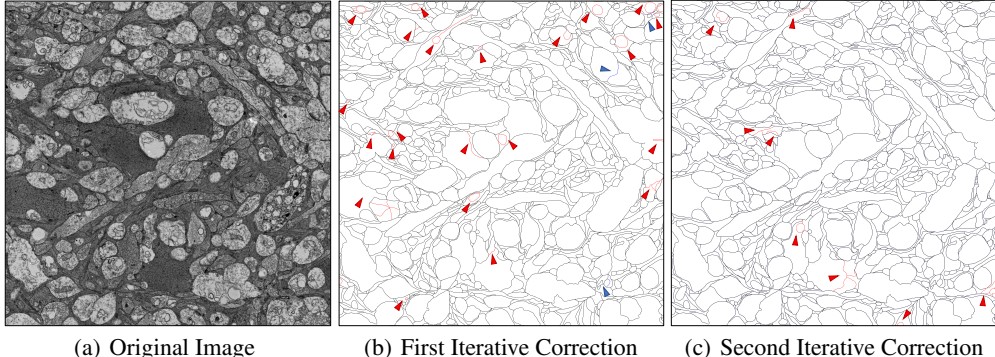

| (a) Original Image | (b) First Iterative Correction | (c) Second Iterative Correction |

Figure 3: Example of iterative labeling. (a) shows the original EM image. (b) and (c) shows the first and second round of annotation processes. The red arrows point to the deleted parts during the inspection processes, and the blue arrows point to the added parts.

## 2.2 TRIPLE LABELING PROCESS

The character of high resolution in TEM image can display a much more detailed sub-cellular structure, which requests more patience to label out the cell (Fig. 2(a)). Besides, the imaging quality can be affected by many factors, such as section thickness or sample staining (Fig. 2(b)). And low imaging quality also requests more labeling efforts. Therefore, increasing labeling efforts is essential to completely annotate U-RISC. To guarantee the labeling accuracy, we set up an iterative correction mechanism in the labeling process (Fig. 3). Before starting the annotation, labeling rules were introduced to all annotators. 58 qualified annotators were allowed to participate in the final labeling process. After the first round annotation, 5 experienced lab staff with sufficient background knowledge were responsible to point out labeling errors pixel by pixel during the second and the third rounds of annotation. Finally, the third round annotation results were regarded as the final "ground truth". And previous two rounds of manual annotations are also saved for later analysis. Fig. 3 shows an example of the two inspection processes. We can see that there are quiet a few mislabeled and missed labeled cell membranes in each round. Therefore, the iterative correction mechanism is very necessary.

## 3 PERCEPTION-BASED EVALUATION

In the analysis of EM data, membrane segmentation is generally an indispensable key step. However, in the field of cell membrane segmentation, most of the previous studies, such as Zhou et al. (2018); Chaurasia & Culurciello (2017); Drozdzal et al. (2016), were not specifically designed for high resolution datasets such as U-RISC. In addition, although many researchers discussed various evaluation criteria for medical and general tasks, few researchers actually incorporate them into the design of the architectures of cell membrane segmentation methods.

By comparing the segmentation results of the popular and classic segmentation methods, we found that the widely used evaluation criteria of segmentation were inconsistent with human perception in some cases, which is further discussed through the perceptual consistency experiment (details in Sec. 3.2). To address this issue, we proposed a new evaluation criterion called Perceptual Hausdorff Distance (PHD). The experimental results showed that it was more consistent with human perception.

## 3.1 INCONSISTENCY BETWEEN EXISTING EVALUATION CRITERIA AND PERCEPTION

Many researchers have proposed various metrics for segmentation evaluation ( Yeghiazaryan & Voiculescu (2018); Niessen et al. (2000); Veltkamp & Hagedoorn (2000); Arganda-Carreras et al. (2015b); Lee et al. (2017)). Some of them, which are the most popular, such as F1 score, Dice Coefficient and IoU ( Sasaki et al. (2007); Dice (1945); Kosub (2019)) are used as the evaluations in most segmentation methods ( Ronneberger et al. (2015); Zhou et al. (2018); Chaurasia & Culurciello (2017); Yu et al. (2017); Chen et al. (2019b)). ISBI2012 cell segmentation challenge used Rand scores (V-Rand and V-Info) ( Arganda-Carreras et al. (2015b)) on thinned membrane for evaluation. Recently, researchers made discussions on various boundary overlap metrics for the

evaluation of medical image segmentation (Yeghiazaryan & Voiculescu (2018)). The most popular evaluation criteria, such as F1 score, are based on the statistics of the degree to which pixels are classified correctly. There are also some metrics designed based on point set distance, such as ASSD ( Yeghiazaryan & Voiculescu (2018)), which is not widely used in recent deep learning researches.

However, quality of segmentation should be judged with respect to the ultimate goal. When we need to use segmentation to reconstruct the whole structure of membranes and connect them, such statistics may not be consistent with human perception in cell membrane segmentation tasks. In the process of segmentation experiments, some interesting phenomena were found. Fig. 4 shows an example of the original image with its manual annotation and segmentation results by two methods GLNet ( Chen et al. (2019b)) and U-Net ( Ronneberger et al. (2015)). The scores indicated that (d) was more similar to (b) than (c).

It should be noted that if these segmentation results are used for reconstructing the structure of cells, the mistakes and loss of structure will be more noticeable when subjects inspect the area surrounded by the red dashed lines in the images, Therefore we consider that (c) is a better prediction, because (d) misses some edges. The reason for the three scores of (c) are lower was that the predicted cell membrane of (c) was thicker than manual labeling. Therefore, it can be inferred that the existing evaluation criteria might not sufficiently robust to variations in the thickness and structures of the membrane, and the evaluation result was more likely inconsistent with human perception.

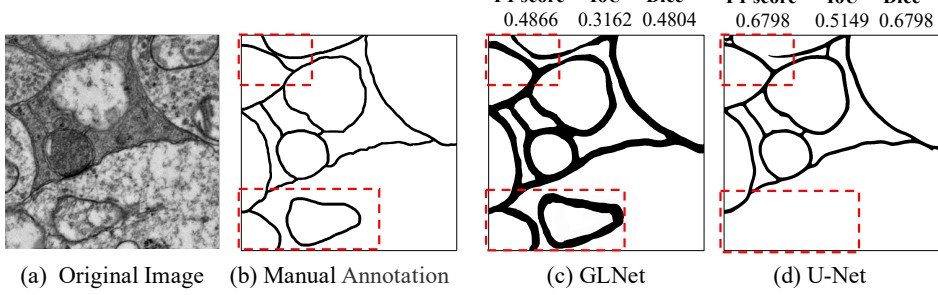

Figure 4: Examples of some segmentation results. (a) is the original EM image. (b) is the manual annotation of cell membrane. (c) and (d) are segmentation results from GLNet and U-Net.

## 3.2 PERCEPTUAL CONSISTENCY EXPERIMENTS

In order to verify the above conjecture, a subjective experiment was designed to explore the consistency with the existing evaluation criteria and human subjective perception. Six popular and classical segmentation methods were used to generate cell membrane segmentation results on U-RISC: U-net ( Ronneberger et al. (2015)), LinkNet( Chaurasia & Culurciello (2017)), CASENet ( Yu et al. (2017)), SENet ( Hu et al. (2018)), U-Net++ ( Zhou et al. (2018)),and GLNet ( Chen et al. (2019b)). Using these segmentation results, 200 groups of images were randomly selected. Each group contained 3 images: the final manual annotation (ground truth) and two automatically generated segmentation results for the same input cell image.

20 subjects were recruited to participate in the experiments. They had either a biological background or experience in cell membrane segmentation and reconstruction. For each group, each of the 20 subjects had three choices. If the subject can tell which segmentation result is more similar to the ground truth, he or she can choose which one. Otherwise, the subject can choose "Difficult to choose". The experiment interface is shown in the Appendix I.

Before the experiment, the subjects were trained on the purpose and source of the images. During the experiment, 200 groups of images were divided into four groups on average in order to prevent the subjects from choosing randomly due to fatigue. For each batch of groups, the subjects needed to complete the judgment continuously without interruption.

After the experiment, for each group, if there were more than 10 votes of the same number, it was called a valid group. Otherwise, it was invalid and discarded. There were a total of 113 valid groups. Then, based on these valid groups, the consistency of the F1 score, IoU, and Dice with human choices was calculated.

According to our experimental results, the consistency of F1 score, IoU, and Dice with human choice was only 34.51%, 35.40%, and 34.51%, respectively. Therefore, it can be inferred that the three criteria are not consistent with human subjective perception in most cases. More results and design of subjective experiments are shown in the Appendix II, and IV,.

### 3.3 PERCEPTUAL HAUSDORFF DISTANCE

Based on the subjective experimental results, it was verified that the widely used evaluation criteria for general segmentation were inconsistent with human perception of cell membrane segmentation. This paper proposes a new evaluation standard based on human perception, namely, **Perceptual Hausdorff Distance (PHD for short)**, considering the structure but ignoring the thickness of cell membrane.

**An Overview of PHD.**

As Fig. 4 shows, from the perspective of neuronal reconstruction, the thickness of the cell membrane is not the key for evaluation. In fact, when the goal is to reconstruct the structure of cells, humans will pay more attention on structure changes, instead of thickness changes. Hence, when measuring the similarity of two cell membrane segmentation results, in order to eliminate the influence of thickness, the segmentation results of two cell membranes were skeletonized, and then the distance between two skeletons was calculated to measure the difference. Since the skeleton is a collection of different points, and Hausdorff distance is a common distance to calculate the difference between two sets of points, the proposed PHD is built upon Hausdorff distance.

On the other hand, through subjective experiments, it was found that people tend to ignore the slight offset between the membrane. Therefore, based on the above two considerations, the Perceptual Hausdorff Distance (PHD) based on Hausdorff distance( Huttenlocher et al. (1993); Aspert et al. (2002); Rachasingho & Tasena (2020)) with modification was designed. Fig. 5 shows the overview of PHD. The details are as follows.

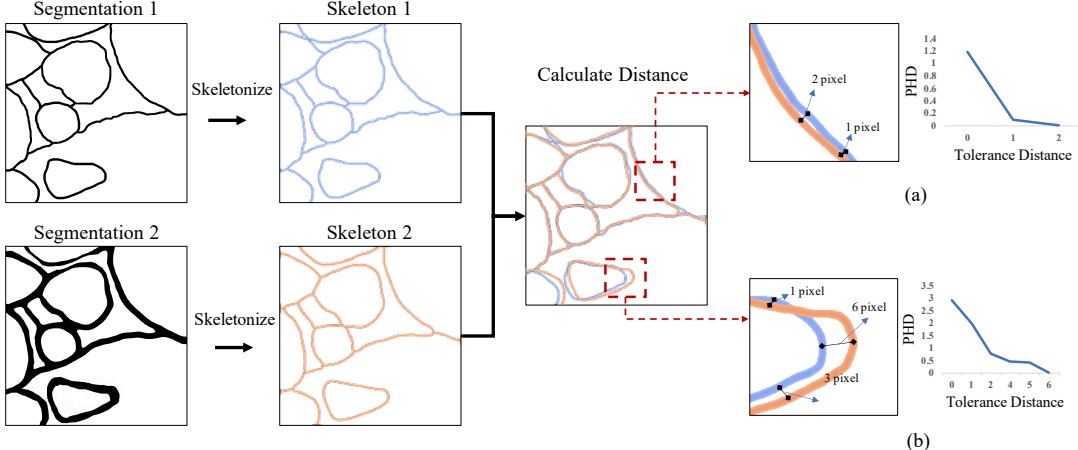

Figure 5: Overview of PHD. The PHD evaluation criterion takes two segmentation results as input. Then, the two inputs are skeletonized. Finally, a distance can be calculated between two skeletons with different tolerance distances. (a) and (b) are two toy cases for intuitively understanding the influence of tolerance distance in PHD.

**Step 1. Skeletonize the cell membrane.** Zhang-Suen thinning algorithm is used ( Zhang & Suen (1984); Saudagar & Mohammed (2016)) and re-implemented to obtain the skeleton of the cell membrane.

**Step 2. Calculate the distance between skeletons.** Hausdorff distance is a common distance used to calculate the difference between two point sets. Consider two unordered nonempty sets of points $\mathbb{X}$ and $\mathbb{Y}$ and the Euclidean distance $d(\boldsymbol{x}, \boldsymbol{y})$ between two point sets. The Hausdorff distance between $\mathbb{X}$ and $\mathbb{Y}$ is defined as

$$d_{\mathrm{H}}(\mathbb{X}, \mathbb{Y}) = \max\left\{d_{\mathbb{X}, \mathbb{Y}}, d_{\mathbb{Y}, \mathbb{X}}\right\} = \max\left\{\max_{\boldsymbol{x} \in \mathbb{X}}\{\min_{\boldsymbol{y} \in \mathbb{Y}} d(\boldsymbol{x}, \boldsymbol{y})\}, \max_{\boldsymbol{y} \in \mathbb{Y}}\{\min_{\boldsymbol{x} \in \mathbb{X}} d(\boldsymbol{x}, \boldsymbol{y})\}\right\},$$

(1)

which can be understood as the maximum value of the shortest distance from a point set to another point set. It is easy to prove that the Hausdorff distance is a metric ( Choi (2019)).

In the task of cell membrane segmentation, we should pay attention to the global distance between two point sets, while Hausdorff distance is sensitive to outliers in two point sets. Therefore, the average distance of the two point sets is obtained naturally by using the average operation instead of all the max operations.

Furthermore, it was found that people have tolerance for the small offset between segmentation results. Specifically, if the distance between two points is very small, people tend to ignore it. Therefore, a concept called **Tolerance Distance** $t$ is defined, which represents human tolerance for small errors.

The Perceptual Hausdorff Distance (PHD) is defined as Eq. 2.

$$d_{\mathrm{PHD}}(\mathbb{X}, \mathbb{Y}) = \frac{1}{|\mathbb{X}|} \sum_{\boldsymbol{x} \in \mathbb{X}} \min_{\boldsymbol{y} \in \mathbb{Y}} d^*(\boldsymbol{x}, \boldsymbol{y}) + \frac{1}{|\mathbb{Y}|} \sum_{\boldsymbol{y} \in \mathbb{Y}} \min_{\boldsymbol{x} \in \mathbb{X}} d^*(\boldsymbol{x}, \boldsymbol{y}), \tag{2}$$

$$d^*(\boldsymbol{x}, \boldsymbol{y}) = \begin{cases} \|\boldsymbol{x} - \boldsymbol{y}\|, & \|\boldsymbol{x} - \boldsymbol{y}\| > t \\ 0, & \|\boldsymbol{x} - \boldsymbol{y}\| \le t \end{cases} \tag{3}$$

To intuitively understand the influence of tolerance distance in PHD, toy cases (a) and (b) as shown in Fig. 5 are taken as examples. In case (a), the blue skeleton scored 19 points while the orange one scored 18 points. Two skeletons are close in the Euclidean Space but do not coincide. Among all the Euclidean distance $d(\boldsymbol{x}, \boldsymbol{y})$ of $\boldsymbol{x} \in \mathbb{X}$ and $\boldsymbol{y} \in \mathbb{Y}$, the max distance is 2 pixels, and the most common distance is 1.

When $t = 0$, which means no mistake can be tolerated, and the PHD is high. If $t = 1$, the PHD value drops a lot. When the $t = 2$, PHD becomes 0. In case (b), there is a large offset between two skeletons. When the $t$ is set to $[2, 4]$, the decline of PHD value is slow. When $t = 6$, it drops to 0, which is the max distance between two point sets of skeletons. The Different settings of $t$ represent the degree of tolerance to the distance between the two skeletons. In practical applications, different tolerance distances can be adopted according to different situations.

**Consistency between PHD and human perception.**

The consistency with human perception of PHD and existing related criteria (TPVF, TNVF, Prec, RVD, Hausdorff, ASSD, V-Rand, and V-Info) based on the subjective experimental results were also calculated (as described in Section 3.2, the formulas can be found in Appendix V). The result showed that compared with other criteria, PHD with appropriate tolerance distance was more consistent with human perception.

As shown in Fig. 6, while tolerance distance t of PHD increasing from 0 to 800, PHD's consistency to human perception rose first and then dropped slowly to 0, suggesting human vision does have tolerance for certain offset. Specifically, the maximum value can be reached at 65.48%, when tolerance distance t was set to 3, suggesting that our perceptions prefer to tolerant small perturbations. It was worthy to note that the optimal PHD score (65.48%) was nearly double of the consistency scores obtained by pixel-error based metrics, such as F1 score.

Our experiment shows that most of these compared criteria in color bars can be improved by skeletonizing the segmentation results before evaluation to a certain extent. In Fig. 6, the consistency of these criteria with human perception calculated based on original images can only reach about 30%, while they can improve about 10% on skeletons. Even the best of these metrics (ASSD) on skeletons can only achieve 52.43%, which is significantly lower than the score of PHD performance with $t = 3$. Therefore, it can be concluded that the PHD performs more consistently with human.

# 4 RE-EXAMINING PHD ON CLASSIC DEEP LEARNING BASED SEGMENTATION METHODS WITH U-RISC

In the previous two sections, we proposed a new ultra-high resolution cell membrane segmentation dataset U-RISC and a new perceptual criteria PHD to help solve the two bottlenecks in the field of cell membrane segmentation. The subjective experiment on a small-scale dataset demonstrated that PHD is more consistent with human perception for the evaluation of cell membrane segmentation than some widely used criteria.

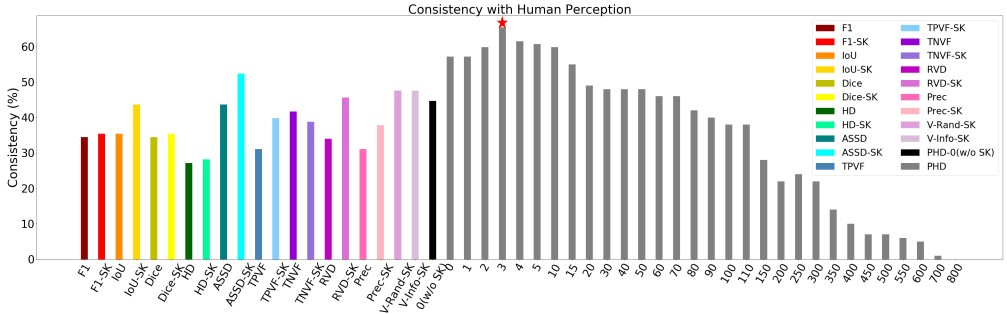

Figure 6: Consistency with human perception. The color bars show the consistency results of F1 score, IoU, TPVF, TNVF, Prec, RVD, Hausdorff, ASSD, V-Rand, and V-Info without and with skeletonization (-SK). The gray bars show the consistency result of PHD. The numbers on X-axis denote the tolerance distance settings.

In order to understand the performance of deep learning methods on the U-RISC dataset, we conducted an in depth investigation on U-RISC with representative deep learning based segmentation methods and different evaluation criteria. To be specific, we chose 6 representative algorithms ( U-net (Ronneberger et al. (2015)), LinkNet(Chaurasia & Culurciello (2017)), CASENet (Yu et al. (2017)), SENet (Hu et al. (2018)), U-Net++ (Zhou et al. (2018)),and GLNet (Chen et al. (2019b))) and re-implemented them on U-RISC dataset. Then four evaluation criteria were used to compare the segmentation results: F1 score, IoU, TPVF, TNVF, Prec, RVD, Hausdorff, ASSD, V-Rand, and V-Info, and PHD.

As mentioned in Sec. 2, the results of the first two rounds of manual labeling results are retained. Therefore, the results of manual annotation under different evaluation criteria will also be analyzed.

**Experiment Settings.** All the six methods use same training data and testing data to compare the performance. And the parameters and loss functions are same as they are proposed in their references. The parameters for each method and other details are shown in Appendix V.

**Experiments Results.** The experiment results were shown in Table. 1. The table showed the scores of different evaluation criteria on the first two rounds of manual annotation results and six segmentation results with the ground truth.

Our first finding was that U-RISC was a challenging dataset in the field of cell membrane segmentation. As shown in Table. 1,the performance of deep learning based methods gained around 0.6 in F1-scores, far below the human level (0.98-0.99) (the first annotation performace) on U-RISC dataset, by contrast, they all exceeded 0.95 on the ISBI 2012. Despite possible improvements by parameter tuning, to such ultra-high resolution images, there was clearly a huge gap between the current popular segmentation methods and human performance.

Our second finding was that evaluation rankings for F1-score, IoU, V-Rand-sk, and V-Info-sk were more consistent with each other, but different from PHD-based rankings. Specifically, while PHD tended to choose CASNet, none of the other metrics chose CASNet as the best choice. According to the subjective experimental results of Sec. 3.2, PHD was much closer to human perception. Therefore, the change of ranking led by PHD may also inspire researchers to re-consider the evaluation criteria for cell membrane segmentation algorithms. It also provides a new perspective for promoting the development of segmentation algorithms.

**Discussion.** Based on the results of these six algorithms, it can be seen that LinkNet and CASENet are better than other methods. From the perspective of network design, LinkNet makes full use of the low-level local information and directly connects the low-level encoder to the decoder of corresponding size. This design pays more attention to the capture of local information, which leads to a more accurate local prediction. CASENet takes full account of the continuity of the edge and makes the low-level features strengthen the high-level semantic information by jumping links between the low-level feature and high-level feature, which pays more attention to structural information. Therefore, the design of LinkNet might be preferred by the traditional evaluation criteria, while CASENet might be preferred by the PHD. This also explains why the two methods rank differently under these two types of evaluation criteria. More local segmentation results of different algorithms are shown in the Appendix III.

Table 1: Experiments on U-RISC Dataset. This table shows the different evaluation results of the first two rounds of human annotations (H.L.1, H.L.2) and six segmentation results: U-Net (Ronneberger et al. (2015)), LinkNet(Chaurasia & Culurciello (2017)), CASENet (Yu et al. (2017)), SENet (Hu et al. (2018)), U-Net++ (Zhou et al. (2018)),and GLNet (Chen et al. (2019b)). PHD-$t$ means the PHD score with tolerance distance $t$. *HD* means Hausdorff. *-sk* means evaluating on skeleton. Note that the ground truth is the third round of human annotation.

| Methods | H.L.1 | H.L.2 | GLNet | CASENet | LinkNet | SENet | U-Net | U-Net++ |
|---|---|---|---|---|---|---|---|---|
| F1↑ | 0.9212 | 0.9933 | 0.4883 | 0.6007 | **0.6070** | 0.5810 | 0.5212 | 0.6030 |
| F1-sk↑ | 0.7958 | 0.9852 | 0.0891 | 0.0995 | **0.1002** | 0.0955 | 0.0871 | 0.1003 |
| IoU ↑ | 0.8616 | 0.9901 | 0.3233 | 0.4307 | **0.4371** | 0.4107 | 0.3541 | 0.4329 |
| IoU-sk ↑ | 0.8760 | 0.9883 | 0.0466 | 0.0524 | 0.0528 | 0.0502 | 0.0456 | **0.0528** |
| Dice ↑ | 0.9212 | 0.9933 | 0.4883 | 0.6007 | **0.6070** | 0.5810 | 0.5212 | 0.6030 |
| Dice-sk ↑ | 0.7958 | 0.9852 | 0.0891 | 0.0995 | 0.4883 | 0.0955 | 0.0871 | **0.1003** |
| V-Rand-sk ↑ | 0.9398 | 0.9910 | 0.4938 | 0.5921 | **0.6310** | 0.5341 | 0.5288 | 0.6211 |
| V-Info-sk ↑ | 0.9410 | 0.9913 | 0.5120 | 0.6013 | **0.6239** | 0.5433 | 0.5178 | 0.6234 |
| TPVF ↑ | 0.9262 | 0.9877 | 0.7657 | 0.8974 | 0.9396 | 0.8970 | 0.5350 | **0.9403** |
| TPVF-sk ↑ | 0.9260 | 0.9770 | 0.1752 | 0.1965 | 0.1971 | 0.1899 | 0.1542 | **0.2016** |
| TNVF ↑ | 0.9993 | 0.9996 | 0.8862 | 0.9622 | 0.9602 | 0.9572 | **0.9768** | 0.9592 |
| TNVF-sk ↑ | 0.9493 | 0.9666 | 0.9950 | 0.9950 | 0.9950 | 0.9949 | **0.9961** | 0.9948 |
| Prec ↑ | 0.9640 | 0.9949 | 0.3524 | 0.5604 | 0.5562 | 0.5339 | **0.5991** | 0.5493 |
| Prec-sk ↑ | 0.8526 | 0.9849 | 0.0914 | 0.1016 | **0.1028** | 0.0971 | 0.1015 | 0.1007 |
| RVD ↓ | 4.3e-06 | 6.6e-08 | 4.0e-04 | 4.0e+07 | 6.0e-05 | 2.0e+08 | 7.8e+08 | **6.4e-05** |
| RVD-sk ↓ | 3.1e-08 | 1.0e-08 | 6.2e+08 | 5.5e+08 | 6.4e+08 | 3.3e+08 | 7.3e+08 | **2.9e+08** |
| ASSD-sk ↓ | 0.1354 | 0.0621 | 1.4970 | 1.2954 | **1.2045** | 1.6575 | 2.0074 | 1.7187 |
| HD-sk ↓ | 3.6441 | 2.0060 | 171.52 | 166.10 | **152.94** | 199.16 | 247.25 | 213.98 |
| **PHD-0** ↓ | 0.2102 | 0.0209 | 1.7646 | 1.7252 | 1.7254 | 1.7404 | 1.7533 | **1.7251** |
| **PHD-1** ↓ | 0.0576 | 0.0145 | 1.6311 | **1.5495** | 1.5508 | 1.5744 | 1.5892 | 1.5535 |
| **PHD-3** ↓ | 0.0456 | 0.0112 | 1.2905 | **1.1345** | 1.1357 | 1.1694 | 1.2009 | 1.1448 |
| **PHD-5** ↓ | 0.0419 | 0.0084 | 1.0018 | 0.8207 | **0.8175** | 0.8643 | 0.9101 | 0.8351 |

In addition, as an example, we add experiments using U-Net, CASENet, and LinkNet on ISBI2012 and SNMI3D datasets (Appendix VI, VIII, IX). The results in Appendix VI show that U-Net with our chosen parameters can perform close to SOTA on ISBI2012 (ours: V-Rand=0.9689,V-Info=0.9723; SOTA: V-Rand=0.9837, V-Info=0.9878 (on skeleton)) and SNEMI3D (ours:V-Rand=0.9389; SOTA: V-Rand=0.9751 (on skeleton)), although we made little efforts in parameter tuning. However, with the same parameter setting, U-Net gets poor scores (V-Rand=0.5288, V-Info=0.5178) on U-RISC. Such a big gap in its performance between U-RISC and previous datasets suggests the challenge from U-RISC dataset, which hopefully will motivate novel designs of machine learning methods in the future. And the results in Appendix VIII and IX show that evaluation rankings for F1-score, IoU, V-Rand-sk, and V-Info-sk are more consistent with each other, but they are different from PHD-based rankings.

## 5  DISCUSSION AND CONCLUSION

This paper aims to solve the two bottlenecks in the development of cell membrane segmentation. Firstly, we proposed U-RISC,Ultra-high Resolution Image Segmentation dataset for Cell membrane, the largest annotated EM dataset for the Cell membrane so far. To our best knowledge, U-RISC is the only uncompressed annotated EM dataset with multiple iterative annotations and uncompressed high-resolution raw image data. During the analysis process of the U-RISC, we found a certain inconsistency between current evaluation criteria for segmentation (e.g. F1 score, IoU) and human perception. Therefore, this article secondly proposed a human-perception based evaluation criterion, called Perceptual Hausdorff Distance (PHD). Through a subjective experiment on a small-scale dataset, experiments results demonstrated that the new criterion is more consistent with human perception for the evaluation of cell membrane segmentation. In addition, the evaluation criteria of PHD and existing classic deep learning segmentation methods are re-examined.

In future research, we will consider how to improve deep learning segmentation methods from the perspective of cell membrane structure and apply PHD criterion for connectomics research. More disccusions are shown in Appendix VII.

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

## A   APPENDIX

**I. Fig. 7 is the interface of perceptual consistency experiments.**

# Which of the left and right pictures on the screen is more similar to the middle picture?

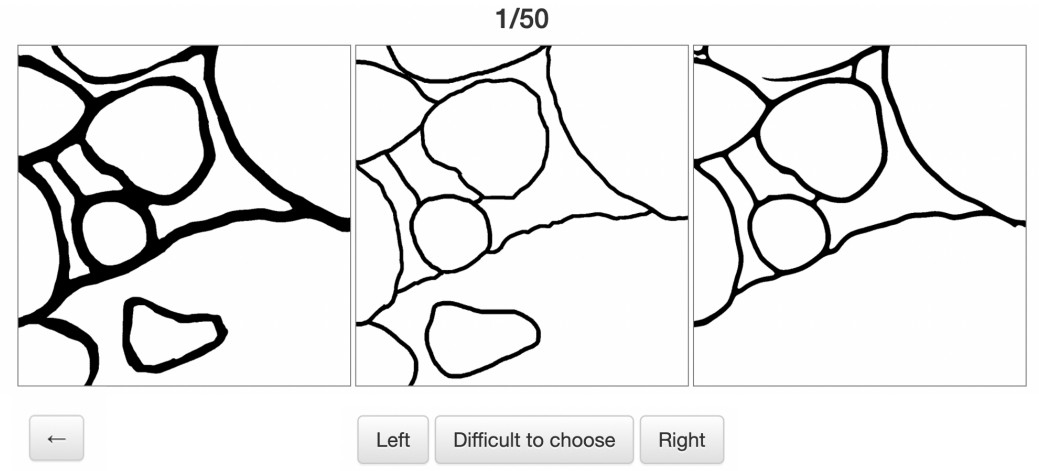

Figure 7: Interface of perceptual consistency experiments.

**II. Fig. 8 and Fig. 9 are some examples of subjective experiment images.**

**III. Fig. 10 and Fig. 11 are some examples of segmentation results of different algorithms.**

**IV. Experiment Details.**

**4.1 Subjective Experiment**

1) Firstly, the 20 human raters were introduced the value of cell membrane segmentation to connectivity and the importance of structure before testing. And then we used several simple examples to teach them about the experimental process.

2) During the formal experiment, the distribution and selection of data were random. The subjects only need to choose one of the two images that they think is more similar to the ground truth.

3) The 200 groups of images for subjective experiment were randomly selected from the segmentation results produced by the above six methods. Therefore, the training data were from the same dataset as those that were used to create the 200 groups of images that the 20 humans evaluated the results.

4) In order to ensure the continuity of the experiment, each subject was asked to judge each group of images within a specified time (less than 10 minutes).

5) In order to prevent the subjects from fatigue in a long period of experiments, 200 groups of images were distributed to the subjects four times on average.

## 4.2 Experiments on U-RISC dataset.

All the six methods use same training data and testing data to compare the performance. And the parameters and loss functions are same as they are proposed in their references.

In the training stage, 60% of the dataset was used as the training data, and then the original image was randomly cut into $1024 \times 1024$ patches to generate 50,000 training images and 20,000 validation images.

Random flipping and clipping were used for data augmentation. Four V100 GPUs were used to train each algorithm. In the testing stage, the original image was cut into the same size of training image, and the patch was tested. These patches were eventually spliced back to the original size for evaluation.

Table 2: Paremeters in methods.

| Paremeters | GLNet | U-Net | SENet | CASENet | U-Net++ | LinkNet |
|---|---|---|---|---|---|---|
| Random Crop | True | True | True | True | True | True |
| Random Crop | True | True | True | True | True | True |
| Learning Rate | 5e-5/2e-5/5e-5 | 1e-3 | 1e-7 | 1e-7 | 1e-7 | 5e-4 |
| Sub Batch Size | 2/1/1 | 4 | 1 | 2 | 1 | 1 |
| Num Epochs | 30/20/30 | 100 | 150 | 300 | 200 | 300 |
| Workers | 12/12/12 | 16 | 16 | 16 | 12 | 8 |

## V. Formulas of criteria mentioned in the texture.

The formulas of metrics we compared are shown in Table 3. The symbols in formulations are explained as follows.

- Precision (also called positive predictive value) is the fraction of relevant instances among the retrieved instances, while recall (also known as sensitivity) is the fraction of the total amount of relevant instances that were actually retrieved.

- TP (true positives), TN (true negatives), FP (false positives), and FN(false negatives) compare the results of the classifier under prediction with ground truth. The terms positive and negative refer to the classifier's prediction, and the terms true and false refer to whether that prediction corresponds to the ground truth.

- $\mathbb{X}$ and $\mathbb{Y}$ are two point sets. $d(\boldsymbol{x}, \boldsymbol{y})$ are the points in $\mathbb{X}$ and $\mathbb{Y}$ respectively.

- In V-Rand (Arganda-Carreras et al. (2015b)), suppose that $S$ is the predicted segmentation and $T$ is the ground truth segmentation. Define $p_{i,j}$ as the probability that a randomly chosen pixel belongs to segment $i$ in $S$ and segment $j$ in $T$. This joint probability distribution satisfies the normalization condition $\sum_{ij} p_{ij} = 1$. The marginal distribution $s_i = \sum_j p_{ij}$ is the probability that a randomly chosen pixel belongs to segment $i$ in $S$, and the marginal distribution $t_j = \sum_i p_{ij}$ is defined similarly.

- In V-Info (Arganda-Carreras et al. (2015b)), the mutual information $I(S;T) = \sum_{ij} p_{ij} \log p_{ij} - \sum_i s_i \log s_i - \sum_j t_j \log t_j$ is a measure of similarity between $S$ and $T$. $H(S) = -\sum_i s_i \log s_i$ is the entropy function.

## VI. Experiments on ISBI2012 and SNEMI3D with U-Net. (Table. 4)

## VII. Further Discussion.

Although the content of this paper is mainly involved in EM cell segmentation, we think the significance is beyond.

Table 3: Formulation of criteria.

| Criteria | Formulation |
|---|---|
| F1 score | $\frac{2 \times precision \times recall}{precision + recall}$ |
| Dice Coefficient | $\frac{2 \times TP}{2 \times TP + FP + FN}$ |
| IoU | $\frac{TP}{TP + FP + FN}$ |
| TPVF | $\frac{TP}{TP + FN}$ |
| TNVF | $\frac{TN}{FP + TN}$ |
| Prec | $\frac{TP}{TP + FP}$ |
| RVD | $\left\| \frac{FP - FN}{TP + FN} \right\|$ |
| Hausdorff | $\max \left\{ \max_{\boldsymbol{x} \in \mathbb{X}} \left\{ \min_{\boldsymbol{y} \in \mathbb{Y}} d(\boldsymbol{x}, \boldsymbol{y}) \right\}, \max_{\boldsymbol{y} \in \mathbb{Y}} \left\{ \min_{\boldsymbol{x} \in \mathbb{X}} d(\boldsymbol{x}, \boldsymbol{y}) \right\} \right\}$ |
| ASSD | $\frac{1}{\|\mathbb{X}\| + \|\mathbb{Y}\|} \left( \sum_{\boldsymbol{x} \in \mathbb{X}} \min_{\boldsymbol{y} \in \mathbb{Y}} d(\boldsymbol{x}, \boldsymbol{y}) + \sum_{\boldsymbol{y} \in \mathbb{Y}} \min_{\boldsymbol{X} \in \mathbb{X}} d(\boldsymbol{x}, \boldsymbol{y}) \right)$ |
| V-Rand | $V_\alpha^{\text{Rand}} = \frac{\sum_{ij} p_{ij}^2}{\alpha \sum_k s_k^2 + (1 - \alpha) \sum_k t_k^2}$ |
| V-Info | $V_\alpha^{\text{info}} = \frac{I(S;T)}{(1 - \alpha) H(S) + \alpha H(T)}$ |
| PHD | $\frac{1}{\|\mathbb{X}\|} \sum_{\boldsymbol{x} \in \mathbb{X}} \min_{\boldsymbol{y} \in \mathbb{Y}} d^*(\boldsymbol{x}, \boldsymbol{y}) + \frac{1}{\|\mathbb{Y}\|} \sum_{\boldsymbol{y} \in \mathbb{Y}} \min_{\boldsymbol{x} \in \mathbb{X}} d^*(\boldsymbol{x}, \boldsymbol{y})$ 
 $d^*(\boldsymbol{x}, \boldsymbol{y}) = \begin{cases} \|\boldsymbol{x} - \boldsymbol{y}\|, & \|\boldsymbol{x} - \boldsymbol{y}\| > t \\ 0, & \|\boldsymbol{x} - \boldsymbol{y}\| \leq t \end{cases}$ |

According to experimental results, the popular methods do not perform well on our dataset (exceeded 95% on ISBI2012, while about 60% on U-RISC). It shows that U-RISC is a challenging dataset that can promote the development of related machine learning and deep learning methods.

The U-RISC dataset may reveal several classic challenges in the field that haven't been solved: One challenge might be the "imbalance problem of samples" ( Alejo et al. (2016); Li et al. (2010); Zhang et al. (2020)). Due to ultra-high resolution images, the pixels of labeled cell membranes only account for 5.64% of total pixels in training sets, in contrast to 21.96% in ISBS2012 and 33.23% in SNEMI3D. The future design of deep learning methods on U-RISC will have to solve this issue.

Some other challenges might include, e.g. ultra high-resolution image segmentation ( Demir et al. (2018); Zhao et al. (2018); Chen et al. (2019a), appropriate loss function design ( Sudre et al. (2017); Spring (1993); Choromanska et al. (2015)),and the issues related to "unclosed" edges as suggested by the reviewer 3.

Taken together, we strongly believe that the U-RISC dataset will have great contribution in technique novelty, by revealing defects in the existing popular methods and promoting novel algorithms for solving classic challenges in machine learning or deep learning community.

In addition, the design of evaluation criteria has been widely concerned in the field of computer science ( Gerl et al. (2020); Lin et al. (2015); Liu et al. (2018)). The PHD we proposed may inspire researchers from a new perspective and further promote the developments of algorithms. The technical novelties of the PHD metric lie in many aspects. To list a few, (1) It can be potentially used

Table 4: Experiments on ISBI2012 and SNEMI3D.

| Methods | F1 ↑ | IoU ↑ | TPVF↑ | TNVF↑ | Prec↑ |
|---------|------|-------|-------|-------|-------|
| ISBI2012 | 0.9701 | 0.9625 | 0.9261 | 0.9461 | 0.9196 |
| SNEMI3D | 0.9380 | 0.9368 | 0.9013 | 0.9324 | 0.8980 |

| Methods | F1-sk ↑ | IoU-sk ↑ | TPVF-sk↑ | TNVF-sk↑ | Prec-sk↑ |
|---------|---------|----------|----------|----------|----------|
| ISBI2012 | 0.3942 | 0.3581 | 0.7097 | 0.9760 | 0.3614 |
| SNEMI3D | 0.2236 | 0.1598 | 0.5466 | 0.9709 | 0.1928 |

| Methods | RVD↓ | ASSD↓ | HD↓ | V-Rand↑ | V-Info↑ |
|---------|------|-------|-----|---------|---------|
| ISBI2012 | 5.7e-05 | 0.3751 | 43.501 | 0.9689 | 0.9723 |
| SNEMI3D | 4.8e-04 | 1.4869 | 108.93 | 0.9389 | 0.9376 |

| Methods | RVD-sk↓ | ASSD-sk↓ | HD-sk↓ | V-Rand-sk↑ | V-Info-sk↑ |
|---------|---------|----------|--------|------------|------------|
| ISBI2012 | 9.3e-06 | 1.1573 | 45.941 | 0.9633 | 0.9601 |
| SNEMI3D | 8.4e-06 | 1.4210 | 130.95 | 0.9201 | 0.9224 |

| Methods | PHD-0 ↓ | PHD-1 ↓ | PHD-3 ↓ | PHD-5 ↓ |
|---------|---------|---------|---------|---------|
| ISBI2012 | 1.0495 | 0.4650 | 0.1043 | 0.0480 |
| SNEMI3D | 1.4012 | 1.0730 | 0.7666 | 0.6124 |

in other tasks, such as vascular segmentation ( Gerl et al. (2020)), bone segmentation ( Lin et al. (2015)), edge detection ( Liu et al. (2018)), and other tasks related to structural and shape information. For example, ( Gerl et al. (2020)) successfully used a distance-based criterion to improve skin layer segmentation in optoacoustic images. (2) It can be modified into loss functions which is also part of our on-going work. It is worthy to note that some works have successfully integrated Hausdorff distance into the loss function ( Genovese et al. (2012); Karimi & Salcudean (2019); Ribera et al. (2019).

**VIII. Experiments on ISBI2012 with U-Net, CASENet, and LinkNet. (Table. 5)**

**IX. Experiments on SNEMI3D with U-Net, CASENet, and LinkNet. (Table. 6)**

Table 5: Experiments on ISBI2012 Datasets. This table shows the different evaluation results of three segmentation results: U-Net (Ronneberger et al. (2015)), CASENet (Yu et al. (2017)), LinkNet(Chaurasia & Culurciello (2017)). PHD-$t$ means the PHD score with tolerance distance $t$. *HD* means Hausdorff. *-sk* means evaluating on skeleton. Note that the ground truth is the third round of human annotation.

| Methods | U-Net | CASENet | LinkNet |
|---|---|---|---|
| F1↑ | 0.9701 | 0.9713 | **0.9724** |
| F1-sk↑ | 0.3942 | **0.4103** | 0.3807 |
| IoU ↑ | 0.9625 | 0.9700 | **0.9701** |
| IoU-sk ↑ | 0.3581 | 0.3377 | **0.3810** |
| V-Rand-sk ↑ | 0.9633 | 0.9653 | **0.9699** |
| V-Info-sk ↑ | 0.9601 | 0.9627 | **0.9656** |
| Dice ↑ | 0.9701 | 0.9713 | **0.9724** |
| Dice-sk ↑ | 0.3942 | **0.4103** | 0.3807 |
| TPVF ↑ | 0.9261 | **0.9326** | 0.9311 |
| TPVF-sk ↑ | **0.7097** | 0.6999 | 0.7036 |
| TNVF ↑ | **0.9461** | 0.9312 | 0.9308 |
| TNVF-sk ↑ | **0.9760** | 0.9532 | 0.9650 |
| Prec ↑ | **0.9196** | 0.9177 | 0.9180 |
| Prec-sk ↑ | **0.3614** | 0.3310 | 0.3198 |
| RVD ↓ | 5.7e-05 | 7.9e-05 | **3.2e-05** |
| RVD-sk ↓ | 9.3e-06 | 1.0e-05 | **4.9e-06** |
| ASSD-sk ↓ | 1.1573 | **0.9211** | 1.0362 |
| HD-sk ↓ | 45.941 | **39.865** | 63.120 |
| **PHD-0** ↓ | 1.0495 | **1.0013** | 1.0235 |
| **PHD-1** ↓ | 0.4650 | **0.3892** | 0.4018 |
| **PHD-3** ↓ | 0.1043 | **0.0997** | 0.1006 |
| **PHD-5** ↓ | 0.0480 | **0.0413** | 0.0438 |

Table 6: Experiments on SNEMI3D Datasets. This table shows the different evaluation results of three segmentation results: U-Net (Ronneberger et al. (2015)), CASENet (Yu et al. (2017)), LinkNet(Chaurasia & Culurciello (2017)). PHD-$t$ means the PHD score with tolerance distance $t$. *HD* means Hausdorff. *-sk* means evaluating on skeleton. Note that the ground truth is the third round of human annotation.

| Methods | U-Net | CASENet | LinkNet |
|---|---|---|---|
| F1↑ | 0.9380 | 0.9389 | **0.9401** |
| F1-sk↑ | 0.2236 | **0.2913** | 0.2830 |
| IoU ↑ | 0.9368 | 0.9373 | **0.9389** |
| IoU-sk ↑ | 0.1598 | 0.1732 | **0.2001** |
| V-Rand-sk ↑ | 0.9201 | 0.9210 | **0.9276** |
| V-Info-sk ↑ | 0.9224 | 0.9226 | **0.9264** |
| Dice ↑ | 0.9380 | 0.9389 | **0.9401** |
| Dice-sk ↑ | 0.2236 | **0.2913** | 0.2830 |
| TPVF ↑ | 0.9013 | 0.9006 | **0.9103** |
| TPVF-sk ↑ | **0.5466** | 0.5349 | 0.5382 |
| TNVF ↑ | **0.9324** | 0.9321 | 0.9299 |
| TNVF-sk ↑ | 0.9709 | 0.9743 | **0.9750** |
| Prec ↑ | 0.8980 | **0.8992** | 0.8793 |
| Prec-sk ↑ | **0.1928** | 0.1917 | 0.1876 |
| RVD ↓ | 4.8e-04 | 4.0e-04 | **8.9e-05** |
| RVD-sk ↓ | 8.4e-06 | 8.0e-06 | **3.6e-06** |
| ASSD-sk ↓ | 1.4210 | 1.3681 | **1.2785** |
| HD-sk ↓ | 130.95 | 156.34 | **129.02** |
| **PHD-0** ↓ | 1.4012 | **1.3792** | 1.3988 |
| **PHD-1** ↓ | 1.0730 | **0.9989** | 1.0215 |
| **PHD-3** ↓ | 0.7666 | **0.6392** | 0.7313 |
| **PHD-5** ↓ | 0.6124 | **0.5821** | 0.5988 |

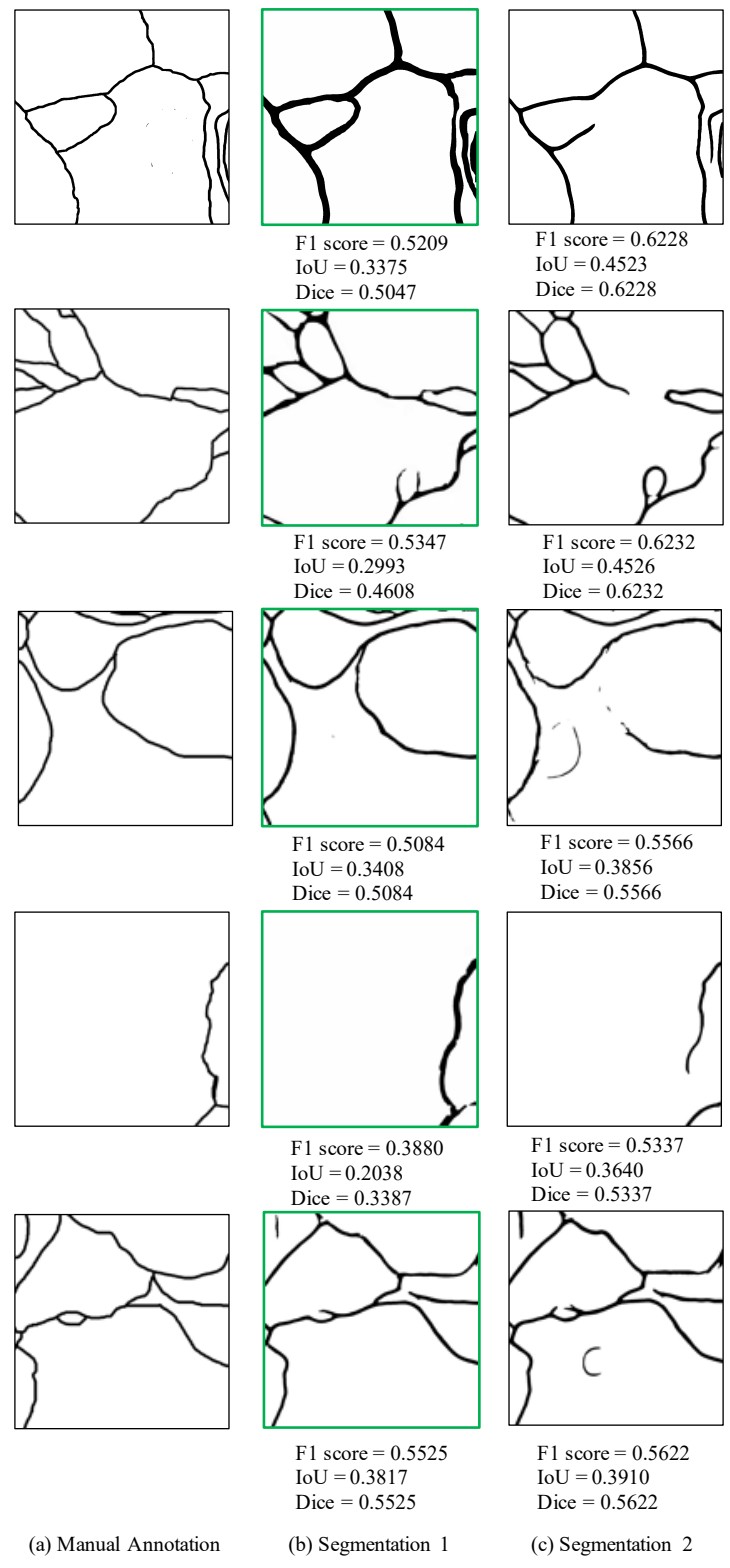

F1 score = 0.5209
IoU = 0.3375
Dice = 0.5047

F1 score = 0.6228
IoU = 0.4523
Dice = 0.6228

F1 score = 0.5347
IoU = 0.2993
Dice = 0.4608

F1 score = 0.6232
IoU = 0.4526
Dice = 0.6232

F1 score = 0.5084
IoU = 0.3408
Dice = 0.5084

F1 score = 0.5566
IoU = 0.3856
Dice = 0.5566

F1 score = 0.3880
IoU = 0.2038
Dice = 0.3387

F1 score = 0.5337
IoU = 0.3640
Dice = 0.5337

F1 score = 0.5525
IoU = 0.3817
Dice = 0.5525

F1 score = 0.5622
IoU = 0.3910
Dice = 0.5622

(a) Manual Annotation     (b) Segmentation 1     (c) Segmentation 2

Figure 8: Examples of subjective experiment images (1). The figure in green box is the choice of most subjects. The scores of F1 score, IOU and Dice are only used to illustrate the inconsistency between the three criteria and human perception, which are not shown to the subjects during the subjective experiments.

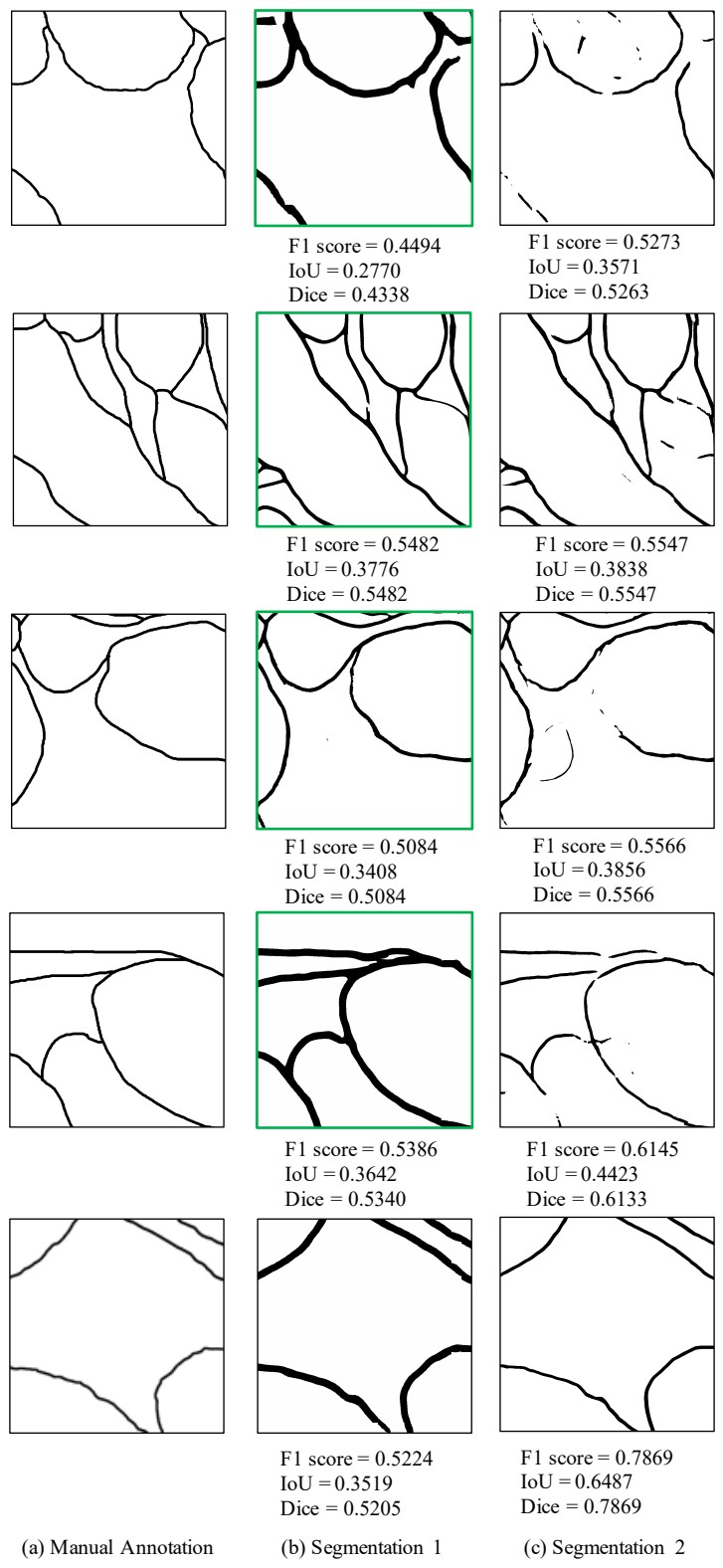

F1 score = 0.4494
IoU = 0.2770
Dice = 0.4338

F1 score = 0.5273
IoU = 0.3571
Dice = 0.5263

F1 score = 0.5482
IoU = 0.3776
Dice = 0.5482

F1 score = 0.5547
IoU = 0.3838
Dice = 0.5547

F1 score = 0.5084
IoU = 0.3408
Dice = 0.5084

F1 score = 0.5566
IoU = 0.3856
Dice = 0.5566

F1 score = 0.5386
IoU = 0.3642
Dice = 0.5340

F1 score = 0.6145
IoU = 0.4423
Dice = 0.6133

F1 score = 0.5224
IoU = 0.3519
Dice = 0.5205

F1 score = 0.7869
IoU = 0.6487
Dice = 0.7869

(a) Manual Annotation     (b) Segmentation  1     (c) Segmentation  2

Figure 9: Examples of subjective experiment images (2). The figure in green box is the choice of most subjects. The scores of F1 score, IOU and Dice are only used to illustrate the inconsistency between the three criteria and human perception, which are not shown to the subjects during the subjective experiments. For the example in the last line, most of the subjects chose "Difficult to choose".

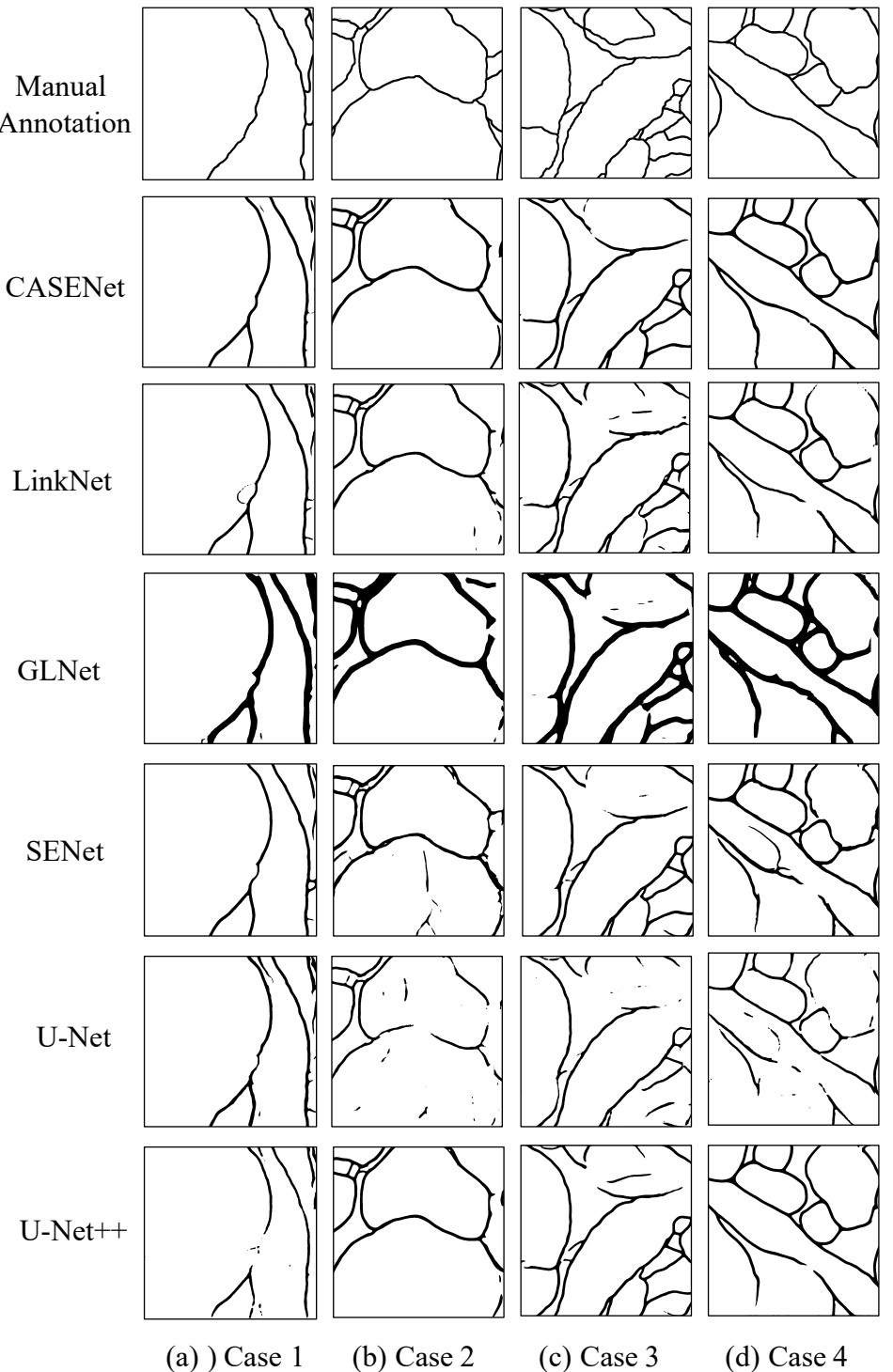

Figure 10: Examples of segmentation results (1).

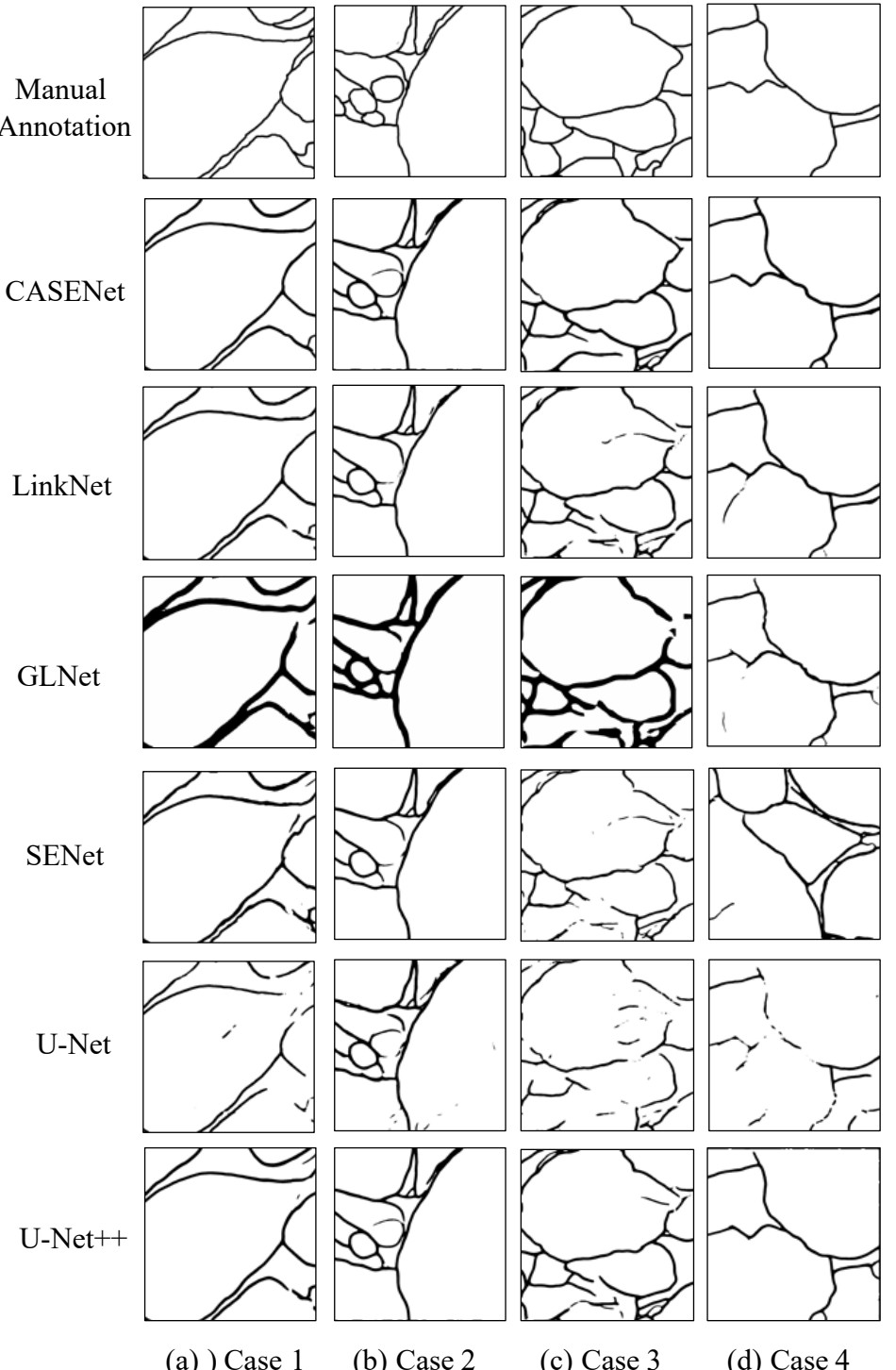

Figure 11: Examples of segmentation results (2).

