# OpenReview forum: "Human Perception-based Evaluation Criterion for Ultra-high Resolution Cell Membrane Segmentation"
_ICLR.cc/2021/Conference — Reject_

### Official Review · AnonReviewer4 · 2020-10-28
**This paper conducted studies on Electron Microscopy image segmentation. The authors built an EM dataset U-RISC with the original high resolution, which is the currently largest annotated EM dataset, and found the current evaluation metrics for membrane segmentation are inconsistent with human perception. The authors therefore proposed a human-perception-based metric, called Perceptual Hausdorff Distance (PHD), which better follows the manual annotation choices than the traditional metrics.**

**Rating:** 4
**Confidence:** 4

**Review:**

Strength: (1) This work proposed an ultra-high-resolution image segmentation dataset for the cell membrane, named U-RISC. The proposed U-RISC is the largest annotated Electron Microscopy (EM) dataset for the cell membrane with multiple iterative annotations and uncompressed high-resolution raw data. Given the uniqueness of the proposed dataset, it is likely to contribute to the future EM based research, such as membrane segmentation.
(2) For membrane segmentation in EM images, the authors develop a human-perception based evaluation criterion, called Perceptual Hausdorff Distance (PHD). Based on the experiments on a small-scale dataset. the proposed PHD metric is better consistency with human perception than the traditional ones.

Weakness: (1) The first concern is on the proposed PHD metrics. The reviewer thinks that there is a lack of comparison analysis between the proposed PHD with the traditional Hausdorff distance. According to Equation 2, the PHD metric is build based on the Hausdorff distance. Therefore, it is necessary to include comparison analysis of using the traditional Hausdorff distance. This will further highlight the novelty of the PHD metric proposed in this paper.
(2) The second concern is the limit technique novelty. The overall contributions of this paper have two parts: establishing a new dataset, and proposing a new PHD metric for the EM membrane segmentation tasks. However, there is no contributions based on the machine learning or deep learning based methods. The reviewer agrees that this manuscript has made some contributions on biomedical image analysis. However, the reviewer thinks this paper cannot meet the requirement of the ICLR conference.
(3) The third concern is the limit validation experiments on the PHD metric. Since the proposed PHD metric is particularly for membrane segmentation, it is supposed to be effective on other EM datasets, such as the ISBI2012 and SNEMI3D challenges. However, the authors did not conduct comparison methods on any other EM datasets. It would be more convincing to conduct experimental analysis on various EM membrane segmentation tasks.

---

> ### Author Response · Authors · 2020-11-18
> **To Reviewer4**
>
> ### Question 3
>
> The third concern is the limit validation experiments on the PHD metric. Since the proposed PHD metric is particularly for membrane segmentation, it is supposed to be effective on other EM datasets, such as the ISBI2012 and SNEMI3D challenges. However, the authors did not conduct comparison methods on any other EM datasets. It would be more convincing to conduct experimental analysis on various EM membrane segmentation tasks.
>
> ### Answer 3
>
> Thank you for the valuable suggestion. We have added experiments on the other two EM segmentation datasets as suggested in Appendix VIII and IX. The results show that evaluation rankings for F1-score, IoU, V-Rand-sk, and V-Info-sk are more consistent with each other, but they are different from PHD-based rankings, which is consistent with previous conclusions.

---

> ### Author Response · Authors · 2020-11-18
> **To Reviewer4**
>
> Thanks for your careful and valuable comments. We will reply your concerns as follows.
>
> ### Question 1
>
> The first concern is on the proposed PHD metrics. The reviewer thinks that there is a lack of comparison analysis between the proposed PHD with the traditional Hausdorff distance. According to Equation 2, the PHD metric is build based on the Hausdorff distance. Therefore, it is necessary to include comparison analysis of using the traditional Hausdorff distance. This will further highlight the novelty of the PHD metric proposed in this paper.
>
> ### Answer 1
>
> Thanks for your valuable suggestion. We have added the traditional Hausdorff distance in the literature review. It is also compared with in two segmentation experiments. The first experiment (Sec.4) used 6 popular methods to train and test on U-RISC dataset, and the second experiment only used U-Net for ISBI2012 and SNEMI3D datasets (Appendix VI). All of the segmentation results were measured by the metrics above. In terms of the consistency with human perception, the traditional Hausdorff distance can only get 27.18%, while PHD is 65.48%. The PHD is significantly higher.
>
> ### Question 2
>
> The second concern is the limit technique novelty. The overall contributions of this paper have two parts: establishing a new dataset, and proposing a new PHD metric for the EM membrane segmentation tasks. However, there is no contributions based on the machine learning or deep learning based methods. The reviewer agrees that this manuscript has made some contributions on biomedical image analysis. However, the reviewer thinks this paper can not meet the requirement of the ICLR conference.
>
> ### Answer 2
>
> Thank you for your concern about the limited technique novelty of our paper. However, we respectfully disagree with this conclusion.
>
> The ISBI2012 and SNEMI3D datasets dominate the evaluation of cell membrane segmentation in EM data, however, the performance of deep learning methods appears to be “saturated” on these datasets (as pointed out by the reviewer 3). Similar concerns were also addressed by KisukLee and his colleagues (Kisuk Lee, et al. NeurIPS, 2017) that “the SNEMI3D challenge has become obsolete in its present form, and must be modified or replaced by a challenge that is capable of properly evaluating algorithms that are now exceedingly accurate.” In our hand, U-net easily exceeded 95% on ISBI2012, but dramatically dropped to about 60% on U-RISC. Such a big gap in performance cannot simply be explained by parameter tuning. In order to improve the performance on U-RISC, more substantial innovations in algorithms are needed.
>
> The U-RISC dataset may reveal several classic challenges that haven’t been solved  in the field: One challenge might be the “imbalance problem of samples” (Alejo R, et al, 2016),( Li D C, et al,2010), ( ZhangK, et al, 2020). Due to ultra-high resolution images, the pixels of labeled cell membranes only account for 5.64% of total pixels in training sets, in contrast to 21.96% in ISBS2012 and 33.23% in SNEMI3D. The future design of deep learning methods on U-RISC will have to solve this issue.
>
> Some other challenges might include, e.g. ultra high-resolution image segmentation (Ilke Demir, et al, 2018), (Hengshuang Zhao, et al, 2018),(Chen W, et al, 2019)) , appropriate loss function design ((SudreC H, et al, 2017), (Spiring F A, et al, 1993), (Choromanska A, et al, 2015)),and the issues related to "unclosed" edges as suggested by the reviewer 3.
>
> Taken together, we strongly believe that the U-RISC dataset will have great contribution in technique novelty, by revealing defects in the existing popular methods and promoting novel algorithms for solving classic challenges in machine learning or deep learning community.
>
> In addition, the design of evaluation criteria has been widely concerned in the field of computer science ((Gerl S, et al, 2020), (Lin P L, et al, 2015), (Liu S,, et al, 2018)). The PHD we proposed may inspire researchers from a new perspective and further promote the developments of algorithms. The technical novelties of the PHD metric lie in many aspects. To list a few, (1) It can be potentially used in other tasks, such as vascular segmentation (Gerl S, et al, 2020), bone segmentation (Lin P L, et al, 2015), edge detection (Liu S, et al, 2018), and other tasks related to structural and shape information. For example, (Gerl S, et al,2020) successfully used a distance-based criterion to improve skin layer segmentation in optoacoustic images. (2) It can be modified into loss functions which is also part of our on-going work. It is worthy to note that some works have successfully integrated Hausdorff distance into the loss function ((Genovese C R, et al, 2012), (Karimi D, et al, 2019), (Ribera J, et al,2019)).

---

### Official Review · AnonReviewer2 · 2020-10-28
**The proposed new public annotated dataset on EM cell membrane segmentation is highly valuable to the community, yet the value of the proposed metric and the presented experiments is questionable**

**Rating:** 3
**Confidence:** 5

**Review:**

# pros:

- To the author's and reviewer's best knowledge, this paper includes the largest annotated public EM data set for cell membrane segmentation (in case it is published with this paper). Until now, the ISBI 2012 challenge (http://brainiac2.mit.edu/isbi_challenge/) dominates the evaluation of cell membrane segmentation in EM data, even though the performance is nearly saturated. New datasets can identify potential weaknesses in similar domains, that are not covered in current datasets and by state of the art methods, yet.

- The discussion about suitable segmentation metrics for cell membrane segmentation is important and must be continued.

- The article is written in a clear and comprehensive manner.


# cons:

- The discussion about appropriate metrics for cell segmentation, that do not depend on the thickness of the segmented cell membrane, has extensively been elaborated in "Crowdsourcing the creation of image segmentation algorithms for connectomics" by Ignacio Arganda-Carreras et al., Frontiers in Neuroanatomy 2015 (9) 142: pp. 1-13. However, this paper is not referenced and the therein proposed metrics are not mentioned or compared.

- The evaluated "state-of-the-art" methods are not "state-of-the-art". They do not correspond to the top entries of the current ISBI Segmentation Challenge Leaderboard. Additionally, no parameters of the methods were adapted.

- It is left unclear how the 20 human raters were instructed to evaluate the segmentation results. For the correct evaluation, not (only) intuitive human perception must be taken into account, but also the usability of the resulting segmentation. The segmentation results on high resolution EM data presented in this paper display many "unclosed" edges, which lead to severe problems, when using the segmentation as a basis for connectivity analysis. To the reviewer's understanding, the proposed Perceptual Hausdorff distance will hardly penalize these errors.

- The dataset is not published in the format of a challenge, which would allow benchmarking on a private test set.

- Spelling should be revised.


# Summary

The presented new high-quality dataset is highly valuabl to the community in order to improve and develop methods for instance segmentation, specifically cell membrane segmentation. The segmentation of thin cell boundaries imposes different challenges and includes different priors, than in other domains of instance segmentation.

The use of appropriate evaluation metrics is crucial to identify suitable und successful methods in experiments and must be critically discussed including domain knowledge. However, in the presented paper, the discussion about suitable metrics is not appropriately linked to the existing literature. A metric is proposed, that is (more) consistent with "human perception". This is an interesting aspect, but its contribution to the successful analysis of neuronal connectivity from EM data remains unclear.

---

> ### Author Response · Authors · 2020-11-18
> **To Reviewer3**
>
> ### Question 3
>
> It is left unclear how the 20 human raters were instructed to evaluate the segmentation results. For the correct evaluation, not (only) intuitive human perception must be taken into account, but also the usability of the resulting segmentation. The segmentation results on high resolution EM data presented in this paper display many "unclosed" edges, which lead to severe problems, when using the segmentation as a basis for connectivity analysis. To the reviewer's understanding, the proposed Perceptual Hausdorff distance will hardly penalize these errors.
>
> ### Answer 3
>
> Thank you for raising this question.
>
> Firstly, the 20 human raters were introduced the value of cell membrane segmentation to connectivity and the importance of structure before testing. And then we used several simple examples to teach them about the experimental process. During the formal experiment, the distribution and selection of data were random. The subjects only need to choose one of the two images that they think is more similar to the ground truth. A detailed introduction has been added in Appendix IV.
>
> Secondly, we are fully aware "closed edges" is important for segmentation and connectivity analysis. In fact, during the labeling and check process of U-RISC, whether the cell membranes were labeled as “closed” edges is one of the major inspection points. Therefore, our dataset itself has very few “unclosed” edges, which in theory should have helped training deep learning methods to avoid "unclosed" edges if the method itself is proper. However, the current tested methods all showed poor performance on U-RISC dataset, suggesting the displayed “unclosed” edges are more likely due to defects of the methods rather than defects of the U-RISC dataset. With the improvement of the performance for deep learning methods, we believe “unclosed” edges in the segmentation results would be greatly reduced.
>
> Although at current stage we didn’t penalize the errors of “unclosed” edges in the proposed PHD, we agree that such penalties might be critical for improving the overall performance of the deep learning methods. However, the remaining questions are, e.g. how much should we set the penalty (as it is a hyperparameter), or should we also set penalties for other bad misalignment errors? In future research, we will keep exploring these questions.
>
> ### Question 4
>
> The dataset is not published in the format of a challenge, which would allow benchmarking on a private test set.
>
> ### Answer 4
>
> Thanks for your valuable consideration. Due to the anonymous requirements of ICLR, we cannot publish the information in the current version. After the acceptance of the paper, we will publish the dataset in the format of a challenge.
>
> ### Question 5
>
> Spelling should be revised.
>
> ### Answer 5
>
> Thanks for your careful reading and suggestions. We have carefully corrected the spelling.

---

> ### Author Response · Authors · 2020-11-18
> **To Reviewer3**
>
> Thanks for your careful and valuable comments. We will reply to your concerns as follows.
>
> ### Question 1
>
> The discussion about appropriate metrics for cell segmentation, that do not depend on the thickness of the segmented cell membrane, has extensively been elaborated in "Crowdsourcing the creation of image segmentation algorithms for connectomics" by Ignacio Arganda-Carreras et al., Frontiers in Neuroanatomy 2015 (9) 142: pp. 1-13. However, this paper is not referenced and the therein proposed metrics are not mentioned or compared.
>
> ### Answer 1
>
> Thanks for your valuable suggestion.
>
> We have added V-Rand and V-Info (on skeleton) (Ignacio Arganda-Carreras et al.) in the literature review of related works. The V-Rand and V-Info and other related metrics were also compared with PHD in the following experiments.
>
> In terms of consistency with human subjective experimental results, the evaluation shows that two newly added metrics, V-Rand and V-Info (on skeleton), both get scores of 47.57%, which is significantly lower than the score of PHD metric (65.48%). Thus, it can be concluded that the PHD performs more consistently with human.
>
> In the segmentation experiment (Sec. 4), we applied 6 popular segmentation methods on U-RISC dataset. All of the segmentation results were measured by the metrics above. The experimental results show that V-Rand, V-Info, F1, Dice, and IOU all choose LinkNet as the best, while PHD tends to have different choices and chooses CASNet. The new results are in line with our previous conclusion, which is "evaluation rankings for F1-score, IoU, and Dice were more consistent with each other, but different from PHD-based rankings."
>
> ### Question 2
>
> The evaluated "state-of-the-art" methods are not "state-of-the-art". They do not correspond to the top entries of the current ISBI Segmentation Challenge Leaderboard. Additionally, no parameters of the methods were adapted.
>
> ### Answer 2
>
> Thank you for raising this question.
>
> Firstly, we have rephrased the statement of "state-of-the-art" as “popular” or “representative”.
>
> Secondly, with regards to the concern that “no parameters of the methods were adapted”: we would like to emphasize that the main purpose of applying these popular methods in this paper is not to design the best method for segmentation. In fact, our purpose is to compare the performance of those popular methods on different datasets and to test if they can equally behave well in the new dataset U-RISC as well as in ISBI2012 or SNEMI3D. Therefore, we do not need to adapt the parameters from the SOTA methods, but only need to remain the same parameter settings for a fair comparison.
>
> In addition, as an example, we add a new experiment using U-Net in ISBI2012 and SNMI3D datasets (Appendix VI). The results show that U-Net with our own parameters can perform close to SOTA on ISBI2012 (ours: V-Rand=0.9689,V-Info=0.9723; SOTA: V-Rand=0.9837, V-Info=0.9878 (on skeleton)) and SNEMI3D (ours: V-Rand=0.9389; SOTA: V-Rand=0.9751 (on skeleton)), although we made little efforts in parameter tuning or data augmentation. However, with the same parameter setting, U-Net only achieves poor scores (V-Rand=0.5288, V-Info=0.5178) in U-RISC. Such a big gap in its performance between U-RISC and previous datasets suggests the challenge from U-RISC dataset, which hopefully will motivate the novel design of machine learning methods in the future.
> The parameters we used for these methods have been added in Appendix IV.

---

### Official Review · AnonReviewer3 · 2020-10-29
**Review - Interesting database but has issues**

**Rating:** 6
**Confidence:** 3

**Review:**

Summary:

This work presents two contributions towards cell membrane segmentation. First, it introduces a new labelled database for this purpose. The authors claim that this is the largest labelled database of high resolution Electron-Microscopy images for this purpose. Second, the work tackles the issue that the F1Score, Dice and IoU scores that evaluate segmentation performance by quantifying the overlap of 2 segmentations are not adequately describing quality of a segmentation with respect to what human experts would prefer for the task. This has been evaluated via employing humans (experts on the task of cell segmentation) to grade which segmentation they prefer, and analysed how their preference correlates with these scores. As a solution, the authors propose a metric, PHD, that can be described as the average Haussdorf distance between the skeletons of two segmentations, together with a threshold tolerance, which they show correlates better with the preference of experts with respect to segmentation quality on this task.

###################################
Reasons for score:

I recommend a rejection of this work for the following reasons. On one hand, constructing a database and releasing it to the community is a great contribution. I am sure that this would be very well accepted. But, on the other hand, I don’t think the article adequately describes the dataset or compare it adequately with existing databases (which makes less of a “database article”). Instead, half the article discusses a metric that is essentially an adaptation of the haussdorf distance (actually, of the “average symmetric surface distance”), adapted in a manner specific to the cell-segmentation task (applied on skeleton, and with a tolerance, the importance of which is questionable). This 2nd contribution has not been accompanied by a literature review on metrics (e.g. only discusses IoU/F1/Dice, missing related distance based metrics like ASSD completely), nor adequately evaluated with such related metrics (besides IoU/Dice/F1). Finally, the modifications, along with many claims in the article, are only relevant to the specific task of cell segmentation (and quite subjective).

###################################

Pros:

1. Great contribution by releasing publicly a new labelled database of high quality. Seems there was a lot of effort to construct good quality ground truth on a number of images much larger than the existing publically available databases. This is definitely interesting for the community that works in this problem.

2. Interesting human-based evaluation of the usefulness of IoU/Dice/F1 scores for the cell-segmentation problem (by 20 humans, this is nice). It is known in the broader community that overlap metrics (IoU/Dice etc) are not perfect, hence there is a lot of work on other metrics [1,2 etc], but this substantiates/quantifies it very nicely.

###################################

Cons:

1. If I would judge the paper focused on the 1st contribution (releasing a database), I would say that it does not contain a sufficient analysis of the database itself, and especially not an adequate description and comparison with other public databases. I think this point could be sufficiently addressed in the rebuttal.

2. From the technical viewpoint, the claimed 2nd contribution is the derivation of a new metric, but the work has not performed any literature review on related work on metrics except IoU/Dice/F1. In fact, the work produces a metric (Eq.2) that seems to me the same as Average Assymetric Surface Distance (ASSD, see [1]), applied to the skeleton (thinned) segmentation, with a tolerance (task-specific modifications). I note that both skeleton-like operations have been previously performed for computing metrics in the cell-segmentation domain (e.g. for evaluation of ISBI2012 challenge: http://brainiac2.mit.edu/isbi_challenge/evaluation, notice the “after thinning” operation). Tolerance-based modifications have also been applied to various other metrics (e.g. [2] below) and are task-specific modifications (and not necessary for the metric to be appropriate in the general sense). In my opinion, this makes the value of the 2nd contribution very low. I think this point cannot be sufficiently addressed in the rebuttal, as I basically think that the contribution of the derived metric is low in comparison to existing literature.

3. The “skeleton” part of PHD is not individually evaluated whether it actually adds substantially. I note that without it, the metric is essentially ASSD with tolerance (tolerance being task-specifically motivated in this context).

4. Evaluation of the proposed metrics is limited, because it does not contain other metrics except IoU/Dice. E.g., it should have been compared with Haussdord, ASSD, etc.

5. The scope of the paper is limited to the cell-segmentation problem.

6. The work contains a number of statements that are not true and would need significant text alterations to reduce them.

The above Cons are described in detail below, with the detailed comments I raise in the “Questions for rebuttal period” section.

References:

Paper with some review on related metrics (there are many more such reviews):
[1] Yeghiazaryan and Voiculescu, Family of boundary overlap metrics for the evaluation of medical image segmentation, Journal of Medical Imaging, 2018

Paper implementing tolerance (which motivates it, and is subjective to the specific task and needs):
[2] Nikolov et al, Deep learning to achieve clinically applicable segmentation of head and neck anatomy for radiotherapy, arxiv 2018


###################################

Questions/points to address during rebuttal period:

The abstract “proposes” a database with “multiple iterative annotations”. However, the actual database released only contains the last 3rd annotation. Please rephrase the abstract so that this is clear to the reader that only 1 annotation should be expected.

Similar to the above, Sec 2.1 claims the existence of multiple (3) annotations as an advantage of the proposed database over other databases (Sec.2.1, “Besides that, U-RICS produced 3 sets of annotations…, all of which can be applied in developing depe learning”). But, these 3 annotations are not released, hence their existence is irrelevant to the reader and this claim/advantage simply does not hold.

The authors claim that the intermediate annotations are “are very valuable for learning” (Sec.2) and therefore will not be released. In this case, if they are valueable, perhaps consider releasing them? Otherwise, I would suggest rephrasing the article, reducing the emphasis on these 3 labels across the whole manuscript, as they are of little relevance to the reader. You can spend the saved space to extend on more related points (e.g. related work etc, see below).

Sec 1. claims: “We found the human performance is far superior to these methods”. I think this statement is very strong and not supported adequately by the current evaluation. I think the authors refer to experiments in Table 1., where the labels from the 1st and 2nd iteration where compared with the results in the 3rd iteration. But, naturally, image after 3rd iteration is conditioned (related) very closely to those from the 1st and 2nd iterations. Of course they will have very high agreement. Also, we note, the 2nd iteration is not a result of 1 human but of multiple (5 experts + annotator). Hence, for these 2 reasons, they cannot support the statement. For a correct assessment for such a claim, segmentations from a single annotator, who is not the same contributing to making the ground truth for the image, should be evaluated against the ground truth. Additionally, inter-rate segmentation performance (multiple humans) could also be evaluated to make such claims. I would recommend this claim to be altered, as well as state explicitly these factors (conditioning of 1st, 2nd, 3rd iterations) in sec 4 to explain how come agreement is so high in Table 1.


The work has performed no literature review on related metrics for segmentation. It discusses and evaluates solely overlap based metrics (Dice/F1 & IoU), and then proposes a distance-based measure. The authors should have discussed and evaluated distance based measures too, and especially have a look at the Average Symmetric Surface Distance (ASSD) metric, which is very similar to what they propose (Eq.2). See [1] as a starting point for related metrics, but there are many more. Many of these papers raise the argument that no metric is enough for all tasks, and that “quality” of a segmentation is subjective to the task. Hence, for each task, one should chose the correct metric, while for “objective” and all-around evaluation of a “general” method (e.g. an arbitrary segmentation network), multiple complimentary metrics should be used. See [2] also with related discussion. Also see that ISBI2012 challenge itself implemented multiple metrics (http://brainiac2.mit.edu/isbi_challenge/evaluation) that are not duscussed here (and their paper also discusses appropriateness of metrics).

Sec.1 “Considering that image compression generally loses many texture details”: Not necessarily. Depends on how much compression, what type of compression, and the actual content. If you would like to claim this, I would suggest you analyse what type of structures disappear if you do a 2x sub-sampling (based on the fact that ISBI has 2x less resolution). Otherwise, I would suggest this is rephrased a bit less strong.


Sec.2.1 intends to do a comparison with ISBI database and SNEMI3D databases. However it still does not provide important information about them. For example, there is no mention of the actual resolution of the other databases, although the work emphasizes a lot in explaining it contains more info due to higher resolution. ISBI seems to be 4 nm x 4 nm x 50 nm / pixel. (from http://brainiac2.mit.edu/isbi_challenge/home), while the introduced one is 2.18 x 2.18 x 70 nm /pixel. Notice that ISBI resolution at z-axis actually is higher than the introduced. This should be made clear in text. Also, please add same information about SNEMI3D. Do you believe this difference in z-axis could make any difference with respect to what structures can be segmented? (In fact, in Sec 2.2, you say that thicker slice affect imaging quality.)

The current database has approximately 2x resolution than ISBI in x,y plane. What type of structures do you believe are not capable of segmenting well in 4x4nm resolution, but capable at 2x2nm resolution? To support the claim that the higher-resolution is a significant advantage (and hence the contribution of releasing such a database is strong), perhaps the work should have performed an evaluation of how useful this extra resolution is in practice, to support the main contirbution.

Please discuss in Sec 2.1 what anatomy are the images of each database coming from. As they are not coming from the same tissue (e.g. this is from retina, ISBI from Larvae cord etc), please discuss if you think this could be a factor for qualitative differences between the databases. If you think it may be, then perhaps claims about what database is more suitable should be adjusted, as perhaps the two have a bit different purpose / characteristics?

Sec. 2.1: “much more challenging”: What evidence is this claim based on? I could make the argument that ISBI may be more challenging to segment due to the lower resolution (hence less information). Please ensure that you back up all claims with appropriate arguments. As it currently stands, this is an unsupported claim and should be removed.

Same comment as above for the “suitable in exploiting cell segmentation algorithms” claim. Why more suitable to exploit algorithms? I think this needs a rephrase.

Sec. 3.1: “may not be consistent with human perception… tasks”, “an natural first instinct was that”, and in Sec 3.3: “humans are more sensitive to structure changes, instead of thickness changes.”. I think these statements are not passing the correct meaning. It is not the “human perception” or the “instinct” of the humans that prioritizes thin (non-)existence of structures over thickness. In your experiments, the evaluators were clearly trained about what the task is. For example, perhaps they know that in cell-segmentation, where the ultimate goal is creating the connectomic, the connectomic can be created regardless the thickness, but a structure should not be missing. Hence, what wrong structures more important than thickness, is the task. Not human perception or instinct (in fact, for me, it’s clearly easier to identify thickness, than locating a small structure missing somewhere in the images). I think these statements pass a wrong meaning. I would suggest that they are rephrased, to emphasize that in every task, where the segmentation itself is not the ultimate goal, the quality of a segmentation should be judged with respect to what the actual ultimate goal is (e.g. here, creating the whole structure of how membranes are connected?). And this should be reflected in the evaluation metrics, where in each task, different metrics, appropriate for the specific one should be used. Please discuss your viewpoint and your recommended amendments.

Same point as the above, in Sec 3.3: “ humans are more sensitive to structure changes, instead of thickness changes.”: I don’t think this statement is in general true. I would say the opposite for me. I can immediately tell that the thickness differs among segmentations, but I have to focus explicitly on certain areas to find whether a specific area has been wrong segmented. I expect the fact that the humans that performed the evaluation were specifically “trained” (Sec 3.2) that they perform *the specific task of cell segmentation* is likely what made them emphasize the actual structure and not care about the thickness. In other words, what criterion/metric is most appropriate has to do with the actual task of interest. Please discuss. I would suggest all related statements about human perception, vision or sensitivity, to be rephrased in a way that is less generic, and instead perhaps passes the message that in each task, quality of segmentation should be judged with respect to the actual ultimate goal.


Sec 3.2 does not describe on what data were the segmentation methods trained. Specifically, the paper should state explicitly if the training data were different images from those that were used to create the 200 groups of images that the 20 humans evaluated the results, or were they the same. Can you please clarify this here and in the text?

Sec 3.1: What is the difference between F1 score and Dice Coefficient? As far as I know (I double checked), these two scores seem the same to me. Phrasing in sec 3.1 suggests they are different. Am I wrong? Please clarify. If they are the same, then perhaps one of them should be removed.

Related to above: Figure4 c: I think that F1 score really is the same as Dice. After you double check, please check this figure. In Fig 4 c, the values of F1/Dice differ. How come? Please double check and clarify. Perhaps implementation detail? Or am I wrong? You see that in the end of Sec 3.2, F1-score and Dice also brought “Exactly” same results for correlation with human perception, agreeing with my view that F1-score/dice are the same. On the other hand, in Table 1, F1 score and Dice differs in *some* methods (humans, GLNet, Unet), but are absolutely the same for other methods (SENet, CASENet, Unet++, LinkNet). Please double check and clarify. I would suggest you check for a small implementation error? Sorry if I misunderstand something, I am happy to hear clarifications.

Sec. 3.3: I think there is no strong technical argument given for introducing tolerance? Without the tolerance, for small offsets/differences, the distance metric will simply have low value. Humans dont "ignore it", it's just small so it does not "bother them enough to mention". Which is exactly what a low value from a distance metric means. From Fig 5, we can see that even without tolerance (=0), the metrics (ASSD on skeleton) behaves perfectly fine, giving higher PHD for the case (b) that has larger distance than case (a). I would recommend adding such explanation and discussion in the paper. What is your view on the above?

Sec 3.3 & Fig 4: “suggesting human vision does have tolerance”: Sure, but this does not necessarily mean it’s the right thing, right? For example, factors for inducing human “tolerance’ can be limited vision capability (our eyes are not as good as a computer in processing pixel-by-pixel) or subjectivity with respect to the task (e.g. if the human annotators know, or they have been trained, that 1-3 pixels is not a important *for the particular task* of cell segmentation). But this does not mean that a metric that has no tolerance is a bad thing, perhaps the metric is even more objective. Please discuss.

“Can F1-score… be improved… refutes this”: This statement is wrong. Skeletonizing *does* improve all these metrics, as shown in Fig 6. They simply don’t reach the result of PHD. Please rephrase.

The evaluation could/should have included other metrics, such as basic Haussdorf distance, ASSD, or metrics used in related challenges, such as Rand etc (see http://brainiac2.mit.edu/isbi_challenge/evaluation)

###################

Minors, or additional feedback for improving the work in the future (not subject to rebuttal):

I would recommend rephrasing the phrase “Surprisingly, we found” in Sec.1, as it is actually a commonly discussed issue in the literature (see my previous comments on related work/references)

Sec.2: “provided by Marc’s lab (Anderson et al. (2011))”: I think this could be rephrased to a more canonical way of refering to a wold, and also be more accurately descriptive? One that clarifies whether the data are exactly those described in Anderson et al 2011? Were they made publically available together with the specific paper (Anderson et al)? Or have they been provided by Marc’s lab (author of the cited work) to you personally for this current work? E.g. a rephrase like “made publically available and described in the work of Anderson et al (2011)” or something like that is more descriptive.

“at an x-y resolution of 2.18 nm/pixel”: is the resolution the same along the 2 axes? If so, clarify something like “2.18nm/pixel across both axes, and 70nm…”

“from different layers”: What is a layer in this context? It has not been defined. Is it a slice? Remember that you are addressing this to a non-domain-specific audience of ICLR, so ensure to be clear about these terms.

Fig 1: “image number” => “number of images” reads better.

Sec. 3.2: “and 2 segmentation results” => 2 “automatically generated” segmentations?

Abstract: “proposes” a dataset? Sounds wrong. I would suggest “introduces” a dataset.

Sec 1: “”how robust if these methods are compared”:  grammar


**================== UPDATE / EDIT AFTER REBUTTAL PERIOD AND UPDATES TO PAPER ========**

Summary of improvements during rebuttal and remaining concerns on the updated manuscript:

Main improvements:

- Improved the comparison with existing EM datasets by extending descriptions and text in Sec 2.1.

- Added an experiment that shows that the same model (Unet) performs significantly better on previous databases than on the proposed (Appendix VIII, IX). This acts as a solid empirical evidence that the current database is indeed more challenging than previous ones, which was previously missing. This supports the value of the database.

- The authors also clarified that they will release all 3 sets of annotations, which can serve various types of methodological developments, as databases like this are not common.

- Added a short discussion of previous work on metrics for segmentation quality, which previously was entirely missing.

- Extended the analysis by incorporating multiple more metrics (~10) that were previously missing, extending significantly over the first version, where only overlap-based metrics were considered (Dice, IoU). The new results support that for the specific problem of cell-segmentaiton, PHD agrees more with human perception than other metrics on this task, including the very related HD and ASSD.

- Showed that the Skeletonization process, part of the proposed metric, improves various metrics (among which the very related HD-based metrics), which fulfills the previous gap of empirical evidence to support its incorporation in the proposed metric.

- Rephrased most points where the text was ambiguous or incorrect.


Summary and Reviewer’s Score adjustment:

Overall, the revision has improved the document significantly. My primary remaining concerns have to do with the actual paper being of interest primarily to the audience interested in the specific task of cell-segmentation, and that the technical value of the PHD metric is relatively limited (as per my initial comments). However, the updated document supports much better the main claims of the paper, and this database could serve as a benchmark for general ML segmentation methods, benefitting the greater ML community. These improvements make me increase my score from a 3 (Clear Reject) to a 6 (Above acceptance threshold).

-- new minor problems in the updated version --

Some new minor points that I noticed. In case the work is accepted, please try to address them for camera ready:

“ASSD… which is not widely used in deep learning researches”: Not a true statement. HD/ASSD/etc are very popular metrics in segmentation tasks (including with DL) and there have been efforts to even turn them into losses. Please rephrase/remove this statement.

“the consistency of the F1 score, IoU and Dice with the human choices was calculated” (Sec. 3.2): Wasn’t this done for the other metrics as well now? Update the text.

Sec.3.2, “Six popular… segmentation results”: Refer readers to appendix for details on training/test config.1?

Sec. 4, “Then four evaluation…”: “four” is not correct after the updates.

Sec. 4, Discussion: “it can be seen… methods”: This no longer holds after the new metrics. Taking into account all metrics, if we naively count for how many metrics a method ranks 1st (which is what the paper did in the first place), then it seems the best is LinkNet, followed by U-net++, not CASENet (which only ranks 1 for PHD-1 and PHD-3).Please update the argument.

Appendix V: “texture” => text

---

> ### Author Response · Authors · 2020-11-20
> **To Reviewer2**
>
> ### Question 5
>
> The scope of the paper is limited to the cell-segmentation problem.
>
> ### Answer 5
>
> Thank you for your concern about the scope of this paper. However, we respectfully disagree with this point.
>
> The ISBI2012 and SNEMI3D datasets dominate the evaluation of cell membrane segmentation in EM data, however, the performance of deep learning methods appears to be “saturated” on these datasets (as pointed out by the reviewer 3). Similar concerns were also addressed by KisukLee and his colleagues (Kisuk Lee, et al. NeurIPS, 2017) that “the SNEMI3D challenge has become obsolete in its present form, and must be modified or replaced by a challenge that is capable of properly evaluating algorithms that are now exceedingly accurate.” In our hand, U-net easily exceeded 95% on ISBI2012, but dramatically dropped to about 60% on U-RISC. Such a big gap in performance cannot simply be explained by parameter tuning. In order to improve the performance on U-RISC, more substantial innovations in algorithms are needed.
>
> The U-RISC dataset may reveal several classic challenges in the field that haven’t been solved:
>
> One challenge might be the “imbalance problem of samples” (Alejo R, et al, 2016), ( Li D C, et al,2010), ( ZhangK, et al, 2020). Due to ultra-high resolution images, the pixels of labeled cell membranes only account for 5.64% of total pixels in training sets, in contrast to 21.96% in ISBS2012 and 33.23% in SNEMI3D. The future design of deep learning methods on U-RISC will have to solve this issue.
>
> Some other challenges might include, e.g. ultra high-resolution image segmentation (Ilke Demir, et al, 2018), (Hengshuang Zhao, et al, 2018),(Chen W, et al, 2019)), appropriate loss function design ((SudreC H, et al, 2017), (Spiring F A, et al, 1993), (Choromanska A, et al, 2015)),and the issues related to "unclosed" edges as suggested by the reviewer 3.
>
> Taken together, we strongly believe that the U-RISC dataset will have great contribution in technique novelty, by revealing defects in the existing popular methods and promoting novel algorithms for solving classic challenges in machine learning or deep learning community.
>
> In addition, the design of evaluation criteria has been widely concerned in the field of computer science ((Gerl S, et al, 2020), (Lin P L, etal, 2015), (Liu S, et al, 2018)). The PHD we proposed may inspire researchers from a new perspective and further promote the developments of algorithms. The technical novelties of the PHD metric lie in many aspects. To listed a few, (1) It can be potentially used in other tasks, such as vascular segmentation (Gerl S, et al, 2020), bone segmentation (Lin P L, etal, 2015), edge detection (Liu S, et al, 2018), and other tasks related to structural and shape information. For example, (Gerl S, et al,2020) successfully used a distance-based criterion to improve skin layer segmentation in optoacoustic images. (2) It can be modified into loss functions which is also part of our on-going work. It is worthy to note that some works have successfully integrated Hausdorff distance into the loss function ((Genovese C R, et al, 2012), (Karimi D, et al, 2019), (Ribera J, et al,2019)).
>
> Therefore, we think the scope of this paper is not limited to the cell-segmentation problem.
>
> ### Question 6
>
> The work contains a number of statements that are not true and would need significant text alterations to reduce them.
>
> ### Answer 6
>
> We are grateful for your careful and valuable comments. We rephrase the statements as shown in Details part and update the new version correspondingly.

---

> > ### Author Response · Authors · 2020-11-20
> > **Details**
> >
> > ### Detail 9
> >
> > “Can F1-score… be improved… refutes this”: This statement is wrong. Skeletonizing does improve all these metrics, as shown in Fig 6. They simply don’t reach the result of PHD. Please rephrase.
> >
> > ### Answer 9
> >
> > Thanks for your suggestion, and we have rephrased the statements as follows.
> >
> > Changes in manuscript:
> >
> >         In Sec. 3.3 (Page 7):
> >
> >         “Can F1 score, IoU, or Dice be improved by skeletonizing the segmentation before evaluation? Our experiment refutes this point of view. In Fig.6, as shown by yellow, lightgreen, and light blue bars, the consistency of the F1 score, IoU, and Dice with human perception calculated based on image skeletons can only reach 44.25\%,44.25\%, 34.51\%, respectively. They are still far from PHD performance with t= 10. Therefore, only extracting image skeleton can not eliminate the shortcomings of existing metrics.”
> >
> >         =>" Our experiment shows that most of these compared criteria in color bars can be improved by skeletonizing the segmentation before evaluation to a certain extent. In Fig. 6, the consistency of these criteria with human perception calculated based on original images can only reach about 30%, while they can improve about 10% on skeletons. Even the best of these metrics (ASSD) on skeletons can only achieve 52.43%, which is significantly lower than the score of PHD performance with t = 10. Therefore, it can be concluded that the PHD performs more consistently with human."
> >
> > ### Detail 10
> >
> > The evaluation could/should have included other metrics, such as basic Haussdorff distance, ASSD, or metrics used in related challenges, such as Rand etc (see http://brainiac2.mit.edu/isbi_challenge/evaluation)
> >
> > ### Answer 10
> >
> > As suggested, we have added these mentioned experiments and discussions in Sec. 3.3 and Sec. 4.
> >
> > In terms of consistency with human subjective experimental results (Sec.3), the evaluation shows that 8 newly added metrics, Hausdorff (27.18%), Hausdorff-SK (28.16%), ASSD (43.69%), ASSD-SK (52.43%), TPVF (31.07%), TPVF-SK (39.81%), TNVF (41.75%,), TNVF-SK (38.83%), RVD (33.98%), RVD-SK (45.63%), Prec (31.07%), Prec-SK (37.86%), V-Rand-SK (47.57%), and V-Info-SK (47.57%), are significantly lower than the score of PHD metric (65.48%). Thus, it can be concluded that The PHD performs more consistently with human perception.
> >
> > In the segmentation experiment on U-RISC (Sec.4), we applied six popular segmentation methods on U-RISC dataset. All of the segmentation results were measured by the metrics above. We found that evaluation rankings for F1-score, IoU, V-Rand-sk, and V-Info-sk were more consistent with each other, but different from PHD-based rankings. Specifically, while PHD tends to choose CASNet, none of the other metrics chose CASNet as the best choice.
> >
> > #### #######
> > Finally, thanks for all your additional feedback and suggestions. We have carefully incorporated them into our paper. Major changes made in the revision are marked in blue so that they can be easily identified.

---

> > ### Author Response · Authors · 2020-11-20
> > **Details**
> >
> > ### Detail 7
> >
> > Sec 3.1: What is the difference between F1 score and Dice Coefficient? As far as I know (I double checked), these two scores seem the same to me. Phrasing in sec 3.1 suggests they are different. Am I wrong? Please clarify. If they are the same, then perhaps one of them should be removed.
> >
> > Related to above: Figure4 c: I think that F1 score really is the same as Dice. After you double check, please check this figure. In Fig 4 c, the values of F1/Dice differ. How come? Please double check and clarify. Perhaps implementation detail? Or am I wrong? You see that in the end of Sec3.2, F1-score and Dice also brought “Exactly” the same results for correlation with human perception, agreeing with my view that F1-score/dice are the same. On the other hand, in Table 1, F1 score and Dice differ in some methods(humans, GLNet, Unet), but are absolutely the same for other methods (SENet, CASENet, Unet++, LinkNet). Please double check and clarify. I would suggest you check for a small implementation error? Sorry if I misunderstand something, I am happy to hear clarifications.
> >
> > ### Answer 7
> >
> > Thanks for your careful reading and raising this issue. In fact, the concept and origin are different for F1 score and Dice Coefficient. The related formulas are shown in Appendix V. Both of the two metrics are used widely for evaluating segmentation algorithms as different metrics (Chicco D, et al.(2020), Z. Zeng, et al. (2019), S. E. AhmedRaza, et al.(2017), (Schwendicke F, et al.(2019), and Zhang D, et al.(2018)). However, in terms of mathematical computation, these two metrics can be the same theoretically. After double check, we found that due to different implementation details these two metrics can have different results, just as you observed and suggested. Therefore, we would like the keep the original results.
> >
> > ### Detail 8
> >
> > Sec. 3.3: I think there is no strong technical argument given for introducing tolerance? Without tolerance, for small offsets/differences, the distance metric will simply have low value. Humans don’t "ignore it", it's just small so it does not "bother them enough to mention". Which is exactly what a low value from a distance metric means. From Fig 5, we can see that even without tolerance (=0), the metrics (ASSD on skeleton) behaves perfectly fine, giving higher PHD for the case (b) that has a larger distance than case (a). I would recommend adding such explanation and discussion in the paper. What is your view on the above?
> >
> > Sec 3.3 & Fig 4: “suggesting human vision does have tolerance”: Sure, but this does not necessarily mean it’s the right thing, right? For example, factors for inducing human “tolerance’ can be limited vision capability (our eyes are not as good as a computer in processing pixel-by-pixel) or subjectivity with respect to the task (e.g. if the human annotators know, or they have been trained, that 1-3 pixels is not important for the particular task of cell segmentation). But this does not mean that a metric that has no tolerance is a bad thing, perhaps the metric is even more objective. Please discuss.
> >
> > ### Answer 8
> >
> > Thanks for your valuable comments.
> >
> > Firstly, with regard to your comments about tolerance, the tolerance in our work is not only a modification, but also a concept. As you mentioned in Question #2, there are also many works introduced the idea of tolerance (Nikolov et al, 2018). The purpose that we introduce the concept of the tolerance is to explore the sensitivity of human perception with respect to the small difference between ground truth and segmentation result as well as its quantization. We agree with that "without tolerance (=0), the metrics behave perfectly fine", however, we think that the exploration of tolerance is still meaningful.
> >
> > Secondly, we agree with you with the comments: “this does not mean that a metric that has no tolerance is a bad thing”. In this paper, the purpose of PHD is to find a better evaluation criterion, at least, in our experiments, it performs better. As for the best evaluation criteria, as you said, it is related to specific tasks, and it cannot be strictly proved by theory.

---

> > ### Author Response · Authors · 2020-11-20
> > **Details**
> >
> > ### Detail 5
> >
> > Sec. 3.1: “may not be consistent with human perception… tasks”, “a natural first instinct was that”, and in Sec 3.3: “humans are more sensitive to structure changes, instead of thickness changes.”. I think these statements are not passing the correct meaning. It is not the “human perception” or the “instinct” of the humans that prioritizes thin (non-)existence of structures over thickness. In your experiments, the evaluators were clearly trained about what the task is. For example, perhaps they know that in cell-segmentation, where the ultimate goal is creating the connectomic, the connectomic can be created regardless the thickness, but a structure should not be missing. Hence, what wrong structures more important than thickness, is the task. Not human perception or instinct (in fact, for me, it’s clearly easier to identify thickness, than locating a small structure missing somewhere in the images). I think these statements pass a wrong meaning. I would suggest that they are rephrased, to emphasize that in every task, where the segmentation itself is not the ultimate goal, the quality of a segmentation should be judged with respect to what the actual ultimate goal is(e.g. here, creating the whole structure of how membranes are connected?). And this should be reflected in the evaluation metrics, where in each task, different metrics, appropriate for the specific one should be used. Please discuss your viewpoint and your recommended amendments.
> >
> > Same point as the above, in Sec 3.3: “humans are more sensitive to structure changes, instead of thickness changes.”: I don’t think this statement in general true. I would say the opposite to me. I can immediately tell that the thickness differs among segmentations, but I have to focus explicitly on certain areas to find whether a specific area has been wrong segmented. I expect the fact that the humans that performed the evaluation were specifically “trained” (Sec 3.2) that they perform the specific task of cell segmentation is likely what made them emphasize the actual structure and not care about the thickness. In other words, what criterion/metric is most appropriate has to do with the actual task of interest. Please discuss. I would suggest all related statements about human perception, vision or sensitivity, to be rephrased in a way that is less generic, and instead perhaps passes the message that in each task, quality of segmentation should be judged with respect to the actual ultimate goal.
> >
> > ### Answer 5
> >
> > We agree with your opinions that “quality of segmentation should be judged with respect to the actual ultimate goal.”, and we have rephrased the related statements as follows.
> >
> > Changes in manuscript:
> >
> >         In Sec. 3.1 (Page 4, Phrase 1): “However, such statistics may not be consistent with human perception in cell membrane segmentation tasks.”
> >
> >         =>“However, quality of segmentation should be judged with respect to the ultimate goal. When we need to use segmentation to reconstruct the whole structure of membranes and connect them, such statistics may not be consistent with human perception of cell membrane segmentation results. There are also some metrics designed based on point set distance, such as ASSD (Yeghiazaryan & Voiculescu (2018)), which is not widely used in recent deep learning researches. ”
> >
> >         In Sec. 3.1 (Page 4, Phrase 1): “...an natural first instinct was that...”
> >
> >         => “...It should be noted that if these segmentation results are used for reconstructing the structure of cells, the mistakes and loss of structure will be more noticeable when subjects inspect the area surrounded by the red dashed lines in the images. Therefore, we consider...”
> >
> >         In Sec. 3.3 (Page 6, Phrase 2): “humans are more sensitive to structure changes, instead of thickness changes.”
> >
> >         => “when the goal is to reconstruct the structure of cells, humans will pay more attention on structure changes, instead of thickness changes.”
> >
> > ### Detail 6
> >
> > Sec 3.2 does not describe on what data were the segmentation methods trained. Specifically, the paper should state explicitly if the training data were different images from those that were used to create the 200 groups of images that the 20 humans evaluated the results, or were they the same. Can you please clarify this here and in the text?
> >
> > ### Answer 6
> >
> > As suggested, we clarify the experiment details in Appendix IV, 4.2. In fact, the training data were from the same dataset as those that were used to create the 200 groups of images that the 20 humans evaluated the results.

---

> > ### Author Response · Authors · 2020-11-20
> > **Details**
> >
> > #### (4)
> >
> > Please discuss in Sec 2.1 what anatomy are the images of each database coming from. As they are not coming from the same tissue (e.g. this is from retina, ISBI from Larvae cord etc), please discuss if you think this could be a factor for qualitative differences between the databases. If you think it maybe, then perhaps claims about what database is more suitable should be adjusted, as perhaps the two have a bit different purpose / characteristics?
> >
> > #### Answer (4)
> >
> > We add the anatomical origin of each dataset in their description accordingly.
> >
> > As you pointed out, the databases came from different tissues, so they have different purpose and characteristics. It’s too subjective to say which one is suitable, we will rephrase their comparison in a more objective manner.
> >
> > #### (5)
> >
> > Sec. 2.1: “much more challenging”: What evidence is this claim based on? I could make the argument that ISBI may be more challenging to segment due to the lower resolution (hence less information). Please ensure that you back up all claims with appropriate arguments. As it currently stands, this is an unsupported claim and should be removed.
> >
> > #### Answer (5)
> >
> > We add the anatomical origin of each dataset in their description accordingly. Again, we feel it is beyond the scope of this paper and our expertise to further discuss details related to anatomical origin.
> >
> > As we explained previously, our dataset might be more challenging than previous datasets, based on the comparison of some popular deep learning methods applied on these datasets.
> >
> > #### (6)
> >
> > Same comment as above for the “suitable in exploiting cell segmentation algorithms” claim. Why more suitable to exploit algorithms? I think this needs a rephrase.
> >
> > #### Answer (6)
> >
> > As we explained previously, our dataset might be more challenging than previous datasets, based on the comparison of some popular deep learning methods applied on these datasets. Meanwhile, it can also be used for some other classic challenges in deep learning, such as imbalance problem of samples (due to much lower ratio of labeled pixels), ultra high-resolution image segmentation, appropriate loss function design and the issues related to "unclosed" edges as suggested by the reviewer 3.

---

> > ### Author Response · Authors · 2020-11-20
> > **Details**
> >
> > ### Detail 4
> >
> > #### (1)
> >
> > Sec.1 “Considering that image compression generally loses many texture details”: Not necessarily. Depends on how much compression, what type of compression, and the actual content. If you would like to claim this, I would suggest you analyse what type of structures disappear if you do a 2x sub-sampling (based on the fact that ISBI has 2x less resolution). Otherwise, I would suggest this is rephrased a bit less strong.
> >
> > #### Answer (1)
> >
> > We agree with your suggestion. Theoretically speaking, 2nm resolution is better than 4nm resolution, but the final image quality actually also depends on tissues, tasks, imaging techniques, and many others. The original statement” Considering…details, how can...original resolution” will be rephrased as
> >
> >         "one risk to these popular and classic methods is that they might be ”saturated” at the current datasets as their performance appear to be ”exceedingly accurate” ( Lee et al. (2017)). How can these classic deep learning based segmentation methods work on new EM datasets with higher resolution and perhaps more challenges? Moreover, how robust of these methods when they are compared with human performance on such EM images?".
> >
> > #### (2)
> >
> > Sec.2.1 intends to do a comparison with ISBI database and SNEMI3Ddatabases. However it still does not provide important information about them. For example, there is no mention of the actual resolution of the other databases, although the work emphasizes a lot in explaining it contains more info due to higher resolution. ISBI seems to be 4 nm x 4 nm x 50 nm / pixel.(from http://brainiac2.mit.edu/isbi_challenge/home), while the introduced one is 2.18 x 2.18 x 70 nm /pixel. Notice that ISBI resolution at z-axis actually is higher than the introduced. This should be made clear in text. Also, please add same information about SNEMI3D. Do you believe this difference in z-axis could make any difference with respect to what structures can be segmented? (In fact, in Sec 2.2, you say that thicker slice affect imaging quality.)
> >
> > #### Answer (2)
> >
> > We will make a more detailed description and comparison between ISBI, SNEMI3D and ours accordingly.
> >
> > Meanwhile, one major difference between previous datasets and the U-RISC might be important for deep learning : the pixels of labeled cell membranes in U-RISC only account for 5.64% of total pixels in the training sets, in contrast to 21.96% in ISBI2012 and 33.23% in SNEMI3D.  The fact that only 5.64% labeled pixels in U-RISC training sets may lead to “imbalance problem of samples” (Alejo R, etal, 2016),( Li D C, et al,2010), ( ZhangK, et al, 2020), a classic challenge in deep learning.
> >
> > However, we think it is beyond the scope of this paper to further discuss what cause such difference in cell membranes/biological samples, because the original data were obtained from a reputable public database and the authors can only ensure the quality of labeling itself.
> >
> > #### (3)
> >
> > The current database has approximately 2x resolution than ISBI in x,y plane. What type of structures do you believe are not capable of segmenting well in 4x4nm resolution, but capable at 2x2nm resolution? To support the claim that the higher-resolution is a significant advantage (and hence the contribution of releasing such a database is strong), perhaps the work should have performed an evaluation of how useful this extra resolution is in practice, to support the main contribution.
> >
> > #### Answer (3)
> >
> > Thank you for raising this concern. We will cautiously claim the advantage of this 2x higher resolution, even if it is intuitively that 2x2 nm resolution is better than 4x4 nm resolution. We have how rephrased our statements in the introduction, as follows:
> >
> >         "one risk to these popular methods is that they might be "saturated" at the current datasets as their performance appear to be "exceedingly accurate" (Kisuk Lee, et al. NeurIPS, 2017). How can these classic deep learning based segmentation methods work on new EM datasets with higher resolution and perhaps more challenges? Moreover, how robust of these methods when they are compared with human performance on such EM images?"
> >
> > However, we do believe the U-RISC dataset is more challenging than previous datasets. To support our argument, we add a new experiment using U-Net in ISBI2012 and SNMI3D datasets (Appendix 6). The results show that U-Net with our own parameters can perform close to SOTA on ISBI2012 (ours: V-Rand=0.9689,V-Info=0.9723; SOTA: V-Rand=0.9837, V-Info=0.9878 (on skeleton)) and SNEMI3D (ours: V-Rand=0.9389; SOTA: V-Rand=0.9751 (on skeleton)), although we made little efforts in parameter tuning or data augmentation. However, with the same parameter setting, U-Net only achieves poor scores (V-Rand=0.5288, V-Info=0.5178) in U-RISC. Such a big gap in its performance between U-RISC and previous datasets suggests the challenge from U-RISC dataset, which hopefully will motivate novel design of machine learning methods in the future.

---

> > ### Author Response · Authors · 2020-11-20
> > **Details**
> >
> > ### Detail 3
> >
> > The work has performed no literature review on related metrics for segmentation. It discusses and evaluates solely overlap based metrics (Dice/F1 & IoU), and then proposes a distance-based measure. The authors should have discussed and evaluated distance-based measures too, and especially have a look at the Average Symmetric Surface Distance (ASSD) metric, which is very similar to what they propose (Eq.2). See [1] as a starting point for related metrics, but there are many more. Many of these papers raise the argument that no metric is enough for all tasks and that “quality” of a segmentation is subjective to the task. Hence, for each task, one should choose the correct metric, while for “objective” and all-around evaluation of a “general” method (e.g. an arbitrary segmentation network), multiple complementary metrics should be used. See [2] also with related discussion. Also, see that ISBI2012 challenge itself implemented multiple metrics (http://brainiac2.mit.edu/isbi_challenge/evaluation) that are not discussed here (and their paper also discusses appropriateness of metrics).
> >
> > ### Answer 3
> >
> > We appreciate your suggestions.
> >
> > Firstly, we have added literature review on related works on metrics in Sec.1 and Sec.3.1.
> >
> > Secondly, we respectfully disagree with the comments "Eq. 2seems the same as ASSD". The PHD metric we proposed and ASSD are different substantially either from the formula itself (Table 3 in Appendix V) or the experimental results, although they are both based on Hausdorff. Mathematically, ASSD computes the average minimum distance of all the points in the two pointsets, while PHD computes the average minimum distance of the points for each point set respectively. Such differences in “average operations” lead to significant different results in experiment results. e.g. in subjective experiments (Fig. 6), ASSD-sk gets 52.43% while PHD-0 gets 57.23%. Not to mention the PHD includes a design of tolerance while ASSD doesn’t (we will further elaborate the significance of this design later).
> >
> > We would like to point out that many popular metrics in the field might look even more “similar” to each other if one can check their formulas in Table 3 in Appendix V. To list a few:
> >
> > IoU   =    TP    /    （TP+FN+FP）
> >
> > Dice  = 2*TP   /   (2*TP+FP+FN)
> >
> > TPVF =   TP    /   （TP+FN）
> >
> > TNVF =   TN   /   （FP+TN）；
> >
> > Prec  =   TP  /    （TP+FP）
> >
> > and so on….
> >
> > Although these metrics may look very “close” to each other, they are not treated as the “same” metric and widely used in the field for various purposes.
> >
> > Thirdly, we agree with you that “no metric is enough for all tasks” and “quality” of segmentation is subjective to the task”. The problem of universality of measurement standards is a problem of great concern in the field. However, in this work, we didn’t emphasize that PHD will be more effective for all tasks. We are not going to propose such a general solution, but try to find a better or suitable one.
> >
> > Finally, for the question for comparing the evaluation metrics in ISBI2012 ( V-Rand, and V-Info). We have added them for evaluation and compared them with PHD, with other related metrics: Hausdorff, ASSD, TNVF, TPVF, Prec, RVD. The related statements and experiments have been added in the comparison of the consistency with human subjective experimental results (Sec.3), segmentation experiments (Sec.4), and segmentation experiments on ISBI2012 and SNEMI3D datasets (Appendix VI).
> >
> > In terms of consistency with human subjective experimental results (Sec.3), the evaluation shows that 8 newly added metrics, Hausdorff (27.18%), HD-SK (28.16%), ASSD (43.69%), ASSD-SK (52.43%), TPVF (31.07%), TPVF-SK (39.81%), TNVF (41.75%,), TNVF-SK (38.83%), RVD (33.98%), RVD-SK (45.63%), Prec (31.07%), Prec-SK (37.86%), V-Rand-SK (47.57%), and V-Info-SK (47.57%), are significantly lower than the score of PHD metric (65.48%). Thus, it can be concluded that The PHD performs more consistently with human perception.
> >
> > In the segmentation experiment on U-RISC (Sec.4), we applied six popular segmentation methods on U-RISC dataset. All of the segmentation results were measured by the metrics above. We found that evaluation rankings for F1-score, IoU, V-Rand-sk, and V-Info-sk were more consistent with each other, but different from PHD-based rankings. Specifically, while PHD tends to choose CASNet, none of the other metrics chose CASNet as the best choice.

---

> > ### Author Response · Authors · 2020-11-20
> > **Details**
> >
> > Now we will explain your detail concerns as below:
> >
> > ### Detail 1
> >
> > The abstract “proposes” a database with “multiple iterative annotations”. However, the actual database released only contains the last 3rd annotation. Please rephrase the abstract so that this is clear to the reader that only 1annotation should be expected.
> >
> > Similar to the above, Sec 2.1 claims the existence of multiple (3) annotations as an advantage of the proposed database over other databases(Sec.2.1, “Besides that, U-RICS produced 3 sets of annotations…, all of which can be applied in developing deep learning”). But, these 3 annotations are not released, hence their existence is irrelevant to the reader and this claim/advantage simply does not hold.
> >
> > The authors claim that the intermediate annotations are “are very valuable for learning” (Sec.2) and therefore will not be released. In this case, if they are valuable, perhaps consider releasing them? Otherwise, I would suggest rephrasing the article, reducing the emphasis on these 3 labels across the whole manuscript, as they are of little relevance to the reader. You can spend the saved space to extend on more related points (e.g. related work etc, see below).
> >
> > ### Answer 1
> >
> > We apologize that our statement "we reserved the intermediate results" (Sec.2) have caused your misunderstanding! In fact, we plan to publish all the results of the three iterative annotations at the same time. To avoid ambiguity, we now rephrase it to "during the relabeling process, we reserved the intermediate results for public release."
> >
> > ### Detail 2
> >
> > Sec 1. claims: “We found the human performance is far superior to these methods”. I think this statement is very strong and not supported adequately by the current evaluation. I think the authors refer to experiments in Table 1., where the labels from the 1st and 2nd iteration were compared with the results in the 3rd iteration. But, naturally, the image after 3rd iteration is conditioned(related) very close to those from the 1st and 2nd iterations. Of course, they will have a very high agreement. Also, we note, the 2nd iteration is not a result of 1 human but of multiple (5 experts + annotator). Hence, for these 2 reasons, they cannot support the statement. For a correct assessment for such a claim, segmentations from a single annotator, who is not the same contributing to making the ground truth for the image, should be evaluated against the ground truth. Additionally, integrate segmentation performance (multiple humans) could also be evaluated to make such claims. I would recommend this claim to be altered, as well as state explicitly these factors (conditioning of 1st, 2nd, 3rd iterations) in sec 4 to explain how come agreement is so high in Table 1.
> >
> > ### Answer 2
> >
> > Firstly, we agree with the comments for Sec. 1, as you suggested, we now have rephrased the related statements as “We found the human performance (measured as the first annotation performance) appears far superior to these methods.” (Sec. 1).
> >
> > Secondly, with respect to the comments for Sec. 4, we have rephrased the statement, as "the performance of deep learning based methods is around 0.6 in terms of F1-scores, far below the human performance which is between 0.98 and 0.99 (the first annotation performance) on U-RISC dataset". However, we would like to mention that the operation of comparing performances of human and machine is not new. Previous works, such as Arganda-Carreras I, et al. (2015) and Lee K, et al. (2017), also report human performance on ISBI2012 and SNEMI3D challenge dataset. Arganda-Carreras I, et al. (2015) reports that human gets V-Rand (on skeleton) scores of 0.999 and 0.999 when first and second labeling on ISBI2012 dataset respectively. And Lee K, et al. (2017) reports that human gets V-Rand score of 0.94.002 on SNEMI3D dataset. Therefore, it wouldn’t be surprised that the first labeling result of human can reach 0.9212 on U-RISC.
> >
> > Additionally, Arganda-Carreras I, et al. (2015) also concluded that “the score of a top algorithm relative to the consensus of two human experts is approaching the score of one human expert relative to the consensus. " Our result indeed supports this opinion. We will add the discussion in our manuscripts to further clarify the experiments.

---

> ### Author Response · Authors · 2020-11-20
> **To Reviewer2**
>
> ### Question 3
>
> The “skeleton” part of PHD is not individually evaluated whether it actually adds substantially. I note that without it, the metric is essentially ASSD with tolerance (tolerance being task-specifically motivated in this context).
>
> ### Answer 3
>
> Thanks for your valuable suggestion.
>
> Firstly, we have computed the consistency of PHD-0 (without skeleton) with human perception (Sec.3.3). The result shows that the performance of PHD-0 (without skeleton) is significantly lower than PHD-0. (PHD-0 (without skeleton) =44.66%, while PHD-0=57.23%)
> Secondly, we have added the comparison of different metrics (Hausdorff, ASSD, TPVF, TNVF, RVD, Prec, V-Rand, and V-Info) about the consistency with human perception (Sec.3). Besides, we also add the scores of metrics for segmentation results in experiments (Sec. 4; Appendix VI). The result shows that without skeleton, PHD-0 and ASSD are also two different metrics (ASSD (on skeleton) =52.43, while PHD-0 = 57.23).
>
> ### Question 4
>
> Evaluation of the proposed metrics is limited, because it does not contain other metrics except IoU/Dice. E.g., it should have been compared with Harsdorff, ASSD, etc.
>
> ### Answer 4
>
> We have added related metrics: Hausdorff, ASSD, TNVF, TPVF, Prec, RVD, V-Rand, and V-Info for evaluation and compared them with PHD. The related statements and experiments have been added in the comparison of the consistency with human subjective experimental results (Sec.3), segmentation experiments (Sec.4), and segmentation experiments on ISBI2012 and SNEMI3D datasets (Appendix VI).
>
> In terms of consistency with human subjective experimental results (Sec.3.3), the evaluation shows that 8 newly added metrics, Hausdorff (27.18%), Hausdorff-SK (28.16%), ASSD (43.69%), ASSD-SK (52.43%), TPVF (31.07%), TPVF-SK (39.81%), TNVF (41.75%,), TNVF-SK (38.83%), RVD (33.98%), RVD-SK (45.63%), Prec (31.07%), Prec-SK (37.86%), V-Rand-SK (47.57%), and V-Info-SK (47.57%), are significantly lower than the score of PHD metric (65.48%). Thus, it can be concluded that The PHD performs more consistently with human perception.
>
> In the segmentation experiment on U-RISC (Sec.4), we applied six popular segmentation methods on U-RISC dataset. All of the segmentation results were measured by the metrics above. We found that evaluation rankings for F1-score, IoU, V-Rand-sk, and V-Info-sk were more consistent with each other, but different from PHD-based rankings. Specifically, while PHD tends to choose CASNet, none of the other metrics chose CASNet as the best choice.

---

> ### Author Response · Authors · 2020-11-20
> **To Reviewer2**
>
> ### Question 2
>
> From the technical viewpoint, the claimed 2nd contribution is the derivation of a new metric, but the work has not performed any literature review on related work on metrics except IoU/Dice/F1. In fact, the work produces a metric (Eq.2) that seems to me the same as Average Asymmetric Surface Distance (ASSD, see [1]), applied to the skeleton (thinned) segmentation, with a tolerance (task-specific codifications).  I note that both skeleton-like operations have been previously performed for computing metrics in the cell-segmentation domain(e.g. for evaluation of ISBI2012 challenge: http://brainiac2.mit.edu/isbi_challenge/evaluation, notice the “after thinning” operation). Tolerance-based modifications have also been applied to various other metrics(e.g. [2] below) and are task-specific modifications (and not necessary for the metric to be appropriate in the general sense). In my opinion, this makes the value of the 2nd contribution very low. I think this point cannot be sufficiently addressed in the rebuttal, as I basically think that the contribution of the derived metric is low in comparison to existing literature.
>
> ### Answer 2
>
> Thanks for your valuable comments.
>
> We have added the literature review on related work on metrics (Hausdorff, ASSD, TPVF, TNVF, RVD, Prec, V-Rand, and V-Info) in Sec.1 and Sec.3.1, including their skeleton-like operations.
>
> However, we respectfully disagree with the comments "Eq. 2seems the same as ASSD". The PHD metric we proposed and ASSD are different substantially either from the formula itself (Table 3 in Appendix V) or the experimental results, although they are both based on Hausdorff. Mathematically, ASSD computes the average minimum distance of all the points in the two pointsets, while PHD computes the average minimum distance of the points for each point set respectively. Such differences in “average operations” lead to significant different results in experiment results. e.g. in subjective experiments (Fig. 6), ASSD-sk gets 52.43% while PHD-0 gets 57.23%. Not to mention the PHD includes a design of tolerance while ASSD doesn’t (we will further elaborate the significance of this design later).
>
> We would like to point out that many popular metrics in the field might look even more “similar” to each other if one can check their formulas in Table 3 in Appendix V. To list a few:
>
> IoU   =    TP    / （ TP + FN+FP）
>
> Dice  = 2*TP   /   (2*TP + FN+FP)
>
> TPVF =  TP    /  （ TP + FN）
>
> TNVF =  TN   /  （ FP + TN )
>
> Prec  =   TP   /  （ TP + FP）
>
> and so on….
>
> Although these metrics may look very “close” to each other, they are not treated as the “same” metric and widely used in the field for various purpose.
>
> Finally, we thank the reviewer for raising the concern regarding the design of “tolerance”, which is critical to understand the novelty of PHD. We agree that similar concepts have been applied to other metrics. The “tolerance” is essentially a hyper-parameter of the metrics, whereas how to find proper hyper-parameters is a long-standing challenge in the machine learning or deep learning community, and sometimes even become a “secret” recipe for a particular task. To the best of our knowledge, there is no single theory can solve this problem.
>
> Alternatively, we speculate that the “tolerance” parameter in the metric is more likely to be subjective than theory-driven. Thus, our major contributions are : (1) we provide an experiment-based evidence that an optimal value does exist in our perception or instinct for tolerance (Fig.6); (2) we further show that humans have the best tolerance for “small” perturbations in shape or structure related tasks, which may help to narrow down the search-space for the future research.
>
> The design of evaluation criteria has been widely concerned in the field of computer science ((Gerl S, et al, 2020), (LinP L, etal, 2015), (Liu S, et al, 2018)). The PHD we proposed and our attempts to explore the tolerance parameter through human-perception perspective may inspire researchers from a new angle and further promote the developments of algorithms.

---

> ### Author Response · Authors · 2020-11-20
> **To Reviewer2**
>
> Thanks for your careful and valuable comments. Firstly, we will explain your 6 major concerns in cons point by point. And the detailed questions will be explained after that. The references we used in the rebuttal can be found in the overall review we added. Major changes made in the revision are marked in blue so that they can be easily identified.
>
> ### Question 1
>
> If I would judge the paper focused on the 1st contribution (releasing a database), I would say that it does not contain a sufficient analysis of the database itself, and especially not an adequate description and comparison with other public databases. I think this point could be sufficiently addressed in the rebuttal.
>
> ### Answer 1
> Thanks for your suggestion. We have added more descriptions and comparisons with other databases in Sec. 2.
>
> Changes in manuscript:
>
>         In Sec. 2 (Page 2,3) (rephrase and add):
>
>         The dataset was annotated upon RC1, a large scale retinal serial section transmission electron microscopic (ssTEM) dataset, publically available upon request and described in the work of Anderson et al. (2011). The original RC1dataset is a 0.25mm diameter, 370 TEM slices volume, spanning the inner nuclear, inner plexiform, and ganglion cell layers, acquired at 2.18 nm/pixel across both axes and 70nm thickness in z-axis. From the 370 serial-section volume, we clipped out 120 images in the size of 10000*10000 pixels from randomly chosen sections. Then, we manually annotated the cell membranes in an iterative annotation-correction procedure.
>
>         ISBI 2012 (Cardona et al. (2010); Arganda-Carreras et al. (2015b); ISBI 2012 (2012)) published a set of 30 images for training, which were captured from the ventral nerve cord of a Drosophila first instar larva at a resolution of 4*4*50 nm/pixel through ssTEM (Arganda-Carreras et al. (2015b); ISBI 2012 (2012)). Each image contains 512*512 pixels, spanning a realistic area of 2*2 μm approximately. In the challenge of SNEMI3D (Kasthuri et al. (2015); ISBI 2013 (2013)), the training data is a 3D stack of 100 images in the size of 1024*1024 pixels with the voxel resolution of 6*6*29nm/pixel. The raw images were acquired at the resolution of 3*3*29 nm/pixel using serial section scanning electron microscopy (ssSEM) from mouses somatosensory cortex (Kasthuri et al. (2015); ISBI 2013 (2013)). U-RISC contains 120 pieces of annotated images (10000*10000 pixels) at the resolution of 2.18*2.18*70 nm/pixel from rabbit retina.
>
>         Due to the difference of species and tissue, U-RISC can fill in the blank of annotated vertebrate retinal segmentation dataset. Besides that, U-RISC has some other characteristics which can be focused on in the future segmentation study. The first one is that the image size and realistic size of URISC is much larger, specifically, the image size of U-RISC is 400 and 100 times of ISBI2012 and SNEMI3D respectively, and the realistic size is 100 and 9 times of them respectively, which can be applied in developing deep learning based segmentation methods according to various demands. And along with the iterative annotation procedure U-RISC actually contains 3 sets of annotation results with increasing accuracy, which could serve as ground truth at different level standard. And the total number of annotated images is 12 and 3.6 times of the public annotated images of ISBI2012 and SNEMI3D respectively (Fig. 1 (d)). An example of the image with its label is shown in the Supplementary. Due to the limitation of the size of the supplementary material, we only uploaded a quarter (5000 * 5000 pixels) size of the original image with its labels.

---

### Official Review · AnonReviewer1 · 2020-10-30
**Good paper**

**Rating:** 7
**Confidence:** 3

**Review:**

This paper presents a large high-resolution cell membrane segmentation dataset and also proposes a new evaluation metric that is more consistent with human perception.  The new metric is called Perceptual Hausdorff Distance (PHD), which first applies thinning to skeletonize the segmentation outcome, then computes the Hausdorff distance between skeletons. PHD has a hyper-parameter, i.e., the tolerance distance, to represent the human's tolerance.


Overall, I think this is a good paper that addresses how to correctly evaluate cell membrane segmentation, which is essential for evaluation at a fair standard but has not been studied extensively. The authors provide strong reasons to illustrate the limitations of existing evaluation metrics,  and present a relatively larger scale high-quality dataset for evaluating different techniques.  The pixel number and the number of images are presented to demonstrate the advantages over ISBI 2012 and SNEMI3D. The image collection, annotation, and evaluation seem to be performed very carefully with 20 subjects involved.


Some minor additional questions:
i) The measurement seems to be only tailored for cell membrane segmentation. Is it possible to make the criterion more generalized?

ii) If I am understanding correctly, the measurement seems very hard to be modified into loss functions since it involves thinning and other heuristics. Is it possible to use PHD not only for evaluation purposes but also for improving standard training?

---

> ### Author Response · Authors · 2020-11-17
> **To Reviewer1**
>
> Thanks for your positive and constructive feedbacks. We will explain your concerns point by point as follows. The references we used in rebuttal can be found in the overall response.
>
> ### Question 1
> The measurement seems to be only tailored for cell membrane segmentation. Is it possible to make the criterion more generalized?
>
> ### Answer 1
> Thank you for your constructive consideration. Although the measurement we proposed is mainly used for cell membrane segmentation, it can be potentially used in other tasks, such as vascular segmentation (Gerl S, et al, 2020), bone segmentation (Lin P L, et al, 2015), edge detection (Liu S, et al, 2018), and other tasks related to structural and shape information. For example, (Gerl S, et al, 2020) successfully used a distance-based criterion to improve skin layer segmentation in optoacoustic images.
>
> ### Question 2
> The measurement seems very hard to be modified into loss functions since it involves thinning and other heuristics. Is it possible to use PHD not only for evaluation purposes but also for improving standard training?
>
> ### Answer 2
> Yes. PHD can be modified into loss functions which is also part of our on-going work. It is worthy to note that some works have successfully integrated Hausdorff distance into loss function ((Genovese C R, et al, 2012), (Karimi D, et al, 2019), (Ribera J, et al, 2019)). Considering the computational complexity caused by thinning, we can use some alternative methods such as efficient skeleton extraction (Au O K C, et al, 2008).

---

### Author Response · Authors · 2020-11-17
**Overall Response**

Dear Reviewers,

We first thank for all the reviewer's timely comments, which are extremely informative and helpful for improving the quality of this paper. In particular, we are glad that the reviewers comment on the proposed dataset as “a great contribution” and “highly valuable to the community”. We have carefully evaluated the reviewers’ comments and thoughtful suggestions responded to these suggestions point by point, and revised the manuscript accordingly.

In addition, we have performed all the experiments/comparisons suggested by the reviewers, and our results remain significant. Therefore, we are confident that the dataset and the PHD metric are important to both neuroscience and machine learning community.

Before the point-to-point responses, we would like to give a summary of our major revisions that have been performed for your quick reference:

  1. add references and discussions for various metrics (Hausdorff, ASSD, TNVF, TPVF, Prec, RVD, V-Rand, and V-Info);
  2. add comparisons for the above mentioned metrics in subjective experiments and final experiments;
  3. add more comparisons for U-RISC and the other two datasets;
  4. add explanations for the design of tolerance;
  5. modify the inaccurate statements in the text.

The references used in the rebuttal can be found here. Our detailed replies to the reviewers’ comments and suggestions can be found below the references.

---

> ### Author Response · Authors · 2020-11-17
> **References2**
>
> [14] Niessen W J, Bouma C J, Vincken K L, et al. Error metrics for quantitative evaluation of medical image segmentation[M] // Performance Characterization in Computer Vision. Springer, Dordrecht, 2000: 275-284.
>
> [15] Veltkamp R C,Hagedoorn M. Shape similarity measures, properties and constructions[C] //International Conference on Advances in Visual Information Systems. Springer, Berlin, Heidelberg, 2000: 467-476.
>
> [16] Arganda-CarrerasI, Turaga S C, Berger D R, et al. Crowdsourcing the creation of image segmentation algorithms for connectomics[J]. Frontiers in neuroanatomy, 2015,9: 142.
>
> [17] Chicco D, Jurman G. The advantages of the Matthews correlation coefficient (MCC) over F1 score and accuracy in binary classification evaluation[J]. BMC genomics, 2020, 21(1):6.
>
> [18] Nikolov S, Blackwell S, Mendes R, et al. Deep learning to achieve clinically applicable segmentation of head and neck anatomy for radiotherapy[J]. arXiv preprintarXiv:1809.04430, 2018.
>
> [19] Z. Zeng, W. Xie, Y. Zhang and Y. Lu, "RIC-Unet: An Improved Neural Network Based on Unet for Nuclei Segmentation in Histology Images," in IEEE Access, vol. 7,pp. 21420-21428, 2019, doi: 10.1109/ACCESS.2019.2896920.
>
> [20] S. E. Ahmed Raza, L. Cheung, D. Epstein, S. Pelengaris, M. Khan and N. M. Rajpoot, "MIMO-Net: A multi-input multi-output convolutional neural network for cell segmentation in fluorescence microscopy images," 2017 IEEE 14thInternational Symposium on Biomedical Imaging (ISBI 2017), Melbourne, VIC,2017, pp. 337-340, doi: 10.1109/ISBI.2017.7950532.
>
> [21] Schwendicke F,Golla T, Dreher M, et al. Convolutional neural networks for dental image diagnostics: A scoping review[J]. Journal of Dentistry, 2019, 91: 103226.
>
> [22] Zhang D, Song Y, Liu D, et al. Panoptic segmentation with an end-to-end cell R-CNN for pathology image analysis[C]//International Conference on Medical Image Computing and Computer-Assisted Intervention. Springer, Cham, 2018: 237-244.
>
> [23] Gerl S, Paetzold J C, He H, et al. A distance-based loss for smooth and continuous skin layer segmentation in optoacoustic images[C] //International Conference on Medical Image Computing and Computer-Assisted Intervention. Springer, Cham, 2020: 309-319.
>
> [24] Lin P L, Huang P W, Huang P Y, et al. Alveolar bone-loss area localization in periodontitis radiographs based on threshold segmentation with a hybrid feature fused of intensity and the H-value of fractional Brownian motion model[J]. Computer methods and programs in biomedicine, 2015, 121(3): 117-126.
>
> [25] Liu S, Ding W, Liu C, et al. ERN: edge loss reinforced semantic segmentation network for remote sensing images[J]. Remote Sensing, 2018, 10(9): 1339.
>
> [26] Au O K C, Tai C L, Chu H K, et al. Skeleton extraction by mesh contraction[J]. ACM transactions on graphics (TOG), 2008, 27(3): 1-10.
>
> [27] Arganda-Carreras I, Turaga S C, Berger D R, et al. Crowdsourcing the creation of image segmentation algorithms for connectomics[J]. Frontiers in neuroanatomy, 2015, 9: 142.
>
> [28] Lee K, Zung J, Li P, et al. Superhuman accuracy on the SNEMI3D connectomics challenge[J]. arXiv preprint arXiv:1706.00120, 2017.

---

> ### Author Response · Authors · 2020-11-17
> **References1**
>
> [1] Alejo R, Monroy-de-Jesús J, Pacheco-Sánchez J H, et al. A selective dynamic sampling back-propagation approach for handling the two-class imbalance problem[J]. Applied Sciences, 2016, 6(7): 200.
>
> [2] Li D C, Liu C W, Hu S C. A learning method for the class imbalance problem with medical datasets [J]. Computers in biology and medicine, 2010, 40(5): 509-518.
>
> [3] Zhang K, Wu Z, Yuan D, et al. Re-weighted Interval Loss for Handling Data Imbalance Problem of End-to-End Keyword Spotting [J]. Proc. Inter speech 2020, 2020: 2567-2571.
>
> [4] Ilke Demir, Krzysztof Koperski, David Lindenbaum, Guan Pang, Jing Huang, Saikat Basu,Forest Hughes, Devis Tuia, and Ramesh Raska. Deepglobe 2018: A challenge to parse the earth through satellite images. In 2018 IEEE/CVF Conference on Computer Vision and Pattern Recognition Workshops (CVPRW), pages 172–17209.IEEE, 2018.
>
> [5] Hengshuang Zhao, Xiaojuan Qi, Xiaoyong Shen, Jianping Shi, and Jiaya Jia. Icnet for real-time semantic segmentation on high-resolution images. In Proceedings of the European Conference on Computer Vision (ECCV), pages 405– 420, 2018.
>
> [6] Chen W, Jiang Z, Wang Z, et al. Collaborative global-local networks for memory-efficient segmentation of ultra-high resolution images[C]//Proceedings of the IEEE Conference on Computer Vision and Pattern Recognition. 2019: 8924-8933. [7] Sudre C H, Li W, Vercauteren T, et al. Generalised dice overlap as a deep learning loss function for highly unbalanced segmentations[M]//Deep learning in medical image analysis and multimodal learning for clinical decision support. Springer, Cham, 2017:240-248.
>
> [8] Spiring F A. The reflected normal loss function[J]. Canadian Journal of Statistics, 1993, 21(3):321-330.
>
> [9] Choromanska A, Henaff M, Mathieu M, et al. The loss surfaces of multilayer networks [C]//Artificial intelligence and statistics. 2015: 192-204.
>
> [10] Genovese C R, Perone-Pacifico M, Verdinelli I, et al. Manifold estimation and singular deconvolution under Hausdorff loss[J]. The Annals of Statistics, 2012, 40(2):941-963.
>
> [11] Karimi D, Salcudean S E. Reducing the hausdorff distance in medical image segmentation with convolutional neural networks[J]. IEEE Transactions on medical imaging, 2019, 39(2): 499-513.
>
> [12] Ribera J, Guera D, Chen Y, et al. Locating objects without bounding boxes [C]//Proceedings of the IEEE Conference on Computer Vision and Pattern Recognition. 2019:6479-6489.
>
> [13] Yeghiazaryan V, Voiculescu I D. Family of boundary overlap metrics for the evaluation of medical image segmentation[J]. Journal of Medical Imaging, 2018, 5(1): 015006.

---

### Decision · Program_Chairs · 2021-01-07
**Final Decision**

**Decision:**

Reject

**Comment:**

This paper focuses on a segmentation of cell imagery (as opposed to the more commonly studied domain of "natural images"). Among its contributions are a novel metric for evaluation of results and a novel dataset. These are acknowledged by the reviewers as strengths. Multiple issues raised in the initial reviews were addressed in the revision (the reviewers agree on this and most of them raised their scores). On the other hand, the concerns remaining have to do with significance and impact. The final evaluation ratings are split, with only a single score clearly in favor of acceptance.

I tend to agree that the contributions, while without a doubt valuable, make this less of a fit to ICLR than to a more specialized venue focusing on biomedical data.